# Bioaccumulation as a driver of high MeHg in the North and Baltic Seas

David J. Amptmeijer[1], Elena Mikheeva[1], Ute Daewel[1], Johannes Bieser[1], and Corina Schrum[1,2]

[1]Matter Transport and Ecosystem Dynamics, Helmholtz-Zentrum Hereon, Geesthacht, Germany
[2]Universität Hamburg, Institute for Marine Sciences, Mittelweg 177, 20146 Hamburg, Germany

**Correspondence:** David J. Amptmeijer (davidamptmeijer@gmail.com)

**Abstract.** Mercury (Hg) is a toxic pollutant that poses significant risks to marine ecosystems and human health as a result of bioaccumulation. Despite its known hazards, the processes that govern Hg bioaccumulation within the marine food web are poorly understood. This study examines the role of the marine ecosystem in Hg cycling in highly productive coastal seas. We integrate Hg biotic uptake, release and transformation into the ECOSMO E2E marine ecosystem model, coupled with the MERCY v2.0 marine Hg cycling model. Incorporating bioaccumulation into the model leads to a 44% increase in total methylmercury (tMeHg) concentrations in coastal pelagic waters, from 0.059 to 0.092 pM, compared to a model without bioaccumulation. Bioaccumulation and binding of Hg to organic matter contribute to elevated Hg levels in surface waters. Furthermore, cyanobacteria-driven reduction of $Hg^{2+}$ to $Hg^0$ decreases average marine Hg concentrations by up to 9% above the mixed layer depth in the Gotland Deep and 20% in shallow Baltic Sea regions, and increases $Hg^0$ evaporation in the Baltic Sea, reducing Hg inflow into the North Sea. We quantify a 1% increase in tMeHg per 4.5 mgC m$^{-3}$ biota biomass. Finally, we show that bioaccumulation decreases the burial of Hg by 13 kg y$^{-1}$ increasing Hg export to the Atlantic Ocean and the English Channel. These findings highlight the importance of ecosystem feedback on marine Hg cycling and demonstrate the need to integrate biological processes into Hg cycling models.

## 1   Introduction

Mercury (Hg) is a naturally occurring toxic element that is extremely persistent in the environment (Driscoll et al., 2013). It can be methylated to methylmercury (MeHg), a dangerous neurotoxin that can bioaccumulate in the marine food chain (Trevors, 1986; Mason et al., 1995). Hg can reach levels $10^6$ times higher in fish than in the surrounding water and can reach levels in seafood that are unsafe for human consumption, especially during pregnancy or for developing children (Hsi et al., 2016; Counter and Buchanan, 2004; Outridge et al., 2018). The toxicity and emission of Hg received worldwide attention in 1956 when Minamata Bay in Japan was polluted with large amounts of Hg. This Hg was methylated by microorganisms and bioaccumulated in the marine ecosystem. Due to the consumption of polluted marine wildlife, more than 1000 people died, and more were permanently disabled (Harada, 1995). Efforts to control Hg emissions culminated in the Minamata Convention on Mercury, which is a pledge to decrease Hg emissions (Outridge et al., 2018). It has 152 parties and is currently signed by 128 countries.

To ensure the effectiveness of the Minamata Convention in controlling the level of Hg pollution, the status of Hg emissions is periodically reviewed in the effectiveness evaluations of the Minamata Convention. In the first draft of the 2023 effectiveness evaluation, it is stated that Hg measurements in the air, biota, and humans are the key parameters to monitor the current risk of Hg (UNEP, 2021). Because samples of high-trophic-level animals are easily available in the form of commercial fish and their Hg levels are comparatively high and thus easier to measure accurately, Hg biomonitoring is often focused on assessing the Hg levels in high trophic levels. In the effectiveness evaluation, it is stated that Hg levels in water can be insightful, but because of the complexity of sampling and the variability in the data, it does not receive the same recommendation for sampling as fish and wildlife receive. While measuring Hg in biota can help evaluate the risk of $MMHg^+$ pollution to humans, and thus support the effective evaluation of the Minamata Convention, fully understanding Hg cycling requires further research. Understanding the link between Hg in the atmosphere and the risk posed to humans by $MMHg^+$ via the consumption of seafood requires studying the factors linking this, including the link between marine Hg cycling and the bioaccumulation of $MMHg^+$ at the base of the food web. Modeling studies are an important tool to improve our understanding of these complex interactions and can help evaluate the effectiveness of Hg reduction strategies.

Because seafood consumption is the dominant source of Hg exposure in humans, due to the formation and subsequent bioaccumulation of MeHg in marine organisms, Hg levels in the world's oceans are of special concern. Oceanic Hg levels are influenced by a variety of sources. Hg can be introduced to the ocean from the atmosphere via atmospheric exchange of $Hg^0$ or by wet deposition of oxidized $Hg^{2+}$. In addition to atmospheric pathways, Hg reaches the global ocean via rivers, sea ice melt, coastal erosion, and hydrothermal vents (Zagar et al., 2006). Hg can be released from the ocean by the evaporation of volatile $Hg^0$ and dimethylmercury (DMHg), or it can be buried in the sediment, removing it from the biosphere (Van Veen et al., 2002; Zagar et al., 2006; Outridge et al., 2018). The dominant species of Hg in surface water is inorganic $Hg^{2+}$. $Hg^{2+}$ and $Hg^0$ are in dynamic redox cycling, but in aquatic environments this favors the oxidized form, $Hg^{2+}$. Although $Hg^0$ can evaporate, $Hg^{2+}$ can be methylated into two forms of organic Hg, monomethylmercury ($MMHg^+$) and DMHg. Both forms are highly toxic, but only DMHg is volatile and can evaporate from surface water, while $MMHg^+$ can bioaccumulate (Morel et al., 1998). The role of DMHg in bioaccumulation is unknown. The sum of $MMHg^+$ and DMHg is referred to as MeHg. In this paper, MeHg refers to all methylated Hg in seawater; this includes both $MMHg^+$ and DMHg. Of these two Hg species, only $MMHg^+$ is known to bioaccumulate. Under anoxic conditions, $Hg^{2+}$ binds with $S^{2-}$ to form cinnabar (HgS), which is considered a sink due to its low solubility (Oliveri et al., 2016). In seawater, the abundance of chloride ions causes $Hg^{2+}$ and $MMHg^+$ to exist mainly in the form of inorganic chloride complexes. The speciation of Hg with organic carbon in the marine ecosystem, such as detritus and DOM, is a complex interaction that can influence the speciation, solubility, mobility, membrane permeability, and toxicity of Hg (Ravichandran, 2004). In this study, we refer to three distinct fractions of both $H^{2+}$ and $MMHg^+$: 1) dissolved species not bound to organic material, including species such as $HgCl_2$ and MMHgCl, collectively referred to as $Hg^{2+}$ and $MMHg^+$, 2) $Hg^{2+}$ and $MMHg^+$ bound to dissolved organic matter (DOM), referred to as Hg-DOM and MMHg-DOM, and $Hg^{2+}$ and $MMHg^+$ bound to detritus, referred to as Hg-detritus and MMHg-detritus.

Bioaccumulation of Hg occurs when uptake exceeds excretion (Bryan, 1979). A key step in bioaccumulation is the uptake of Hg directly from the water through respiration, absorption, or swallowing (Lee and Fisher, 2016). This can lead to a concentra-

tion of Hg inside organisms up to 100,000 times higher than ambient concentration. This process, called **bioconcentration**, is especially important at the base of the food web and can be measured in laboratory studies as the equilibrium between dissolved Hg and concentrations in biota. Different chemical forms, such as $Hg^{2+}$ and $MMHg^+$, can bioconcentrate simultaneously but at different rates (Mason et al., 1996). The second process is the increase of Hg with increasing trophic position, which is called **biomagnification**. As a result of biomagnification, already high bioconcentrated Hg values in phytoplankton can be amplified to dangerous levels in high trophic animals, such as predatory fish, marine mammals, and seabirds (Lavoie et al., 2013). Biomagnification is more variable, as it depends on trophic transfer efficiency and excretion rates. These processes are influenced by multiple factors, including life cycle, water temperature, and diet (Borgå et al., 2004). Biomagnification can be estimated in nature by measuring stable carbon and nitrogen isotopes with Hg to assess both the Hg content and the trophic position of a series of animals (Lavoie et al., 2013). The biomagnification factor is calculated as the rate of increase in Hg per unit increase in trophic position. (Mackay and Fraser, 2000). These processes differ depending on the chemical form of Hg, which will be discussed in more detail below.

As mentioned before, the dominant form of $Hg^{2+}$ and $MMHg^+$ in the marine environments is $HgCl_2$ and $MMHgCl$ respectively. These compounds can diffuse through cell membranes due to their lipophilic nature or bind to organic matter (Zhong and Wang, 2009). This diffusion is mainly dependent on the surface area of organic membranes that are in contact with water and is therefore dominated by microorganisms such as phytoplankton (Mason et al., 1996). Recent work has expanded on this basic understanding of bioaccumulation and has shown that while these lipophilic compounds can diffuse through the cell membrane, total uptake into phytoplankton is a complex two-step process in which Hg binds first to the phycosphere before it is absorbed into the cell. Recent data suggest that $MMHg^+$ uptake is influenced by cell-dependent factors such as phycosphere thickness and availability of transmembrane channels for $MMHg^+$ transport, while this is not the case for $Hg^{2+}$ (Garcia-Arevalo et al., 2024). This suggests that $Hg^{2+}$ only bioaccumulates due to its lipophilic nature, whereas $MMHg^+$ both bioaccumulates due to its lipophilic nature and is actively transported by the cell.

When phytoplankton is grazed, protein-bound $MMHg^+$ uptake is much more efficient than lipid-bound $Hg^{2+}$. This leads to an increased biomagnification factor for $MMHg^+$ compared to $Hg^{2+}$ (Mason et al., 1996). This results in much higher levels of $MMHg^+$ in high trophic animals compared to $Hg^{2+}$, although the concentration of $Hg^{2+}$ is generally an order of magnitude higher than the concentration of $MMHg^+$ (Bieser et al., 2023; Cossa et al., 2017).

In addition to $Hg^{2+}$ and $MMHg^+$ uptake, another role for phytoplankton in Hg cycling is demonstrated by Kuss et al. (2015). Their research showed that certain genera of cyanobacteria in the Baltic Sea (notable Synechococcus and Aphanizomenon) can also react with Hg by reducing $Hg^{2+}$ to $Hg^0$. Since $Hg^0$ is volatile and can evaporate, increasing the fraction of $Hg^0$ can reduce the total concentration of marine Hg by increasing evaporation. The process of biotically induced reduction of $Hg^{2+}$ to $Hg^0$ is referred to as biogenic reduction.

Marine Hg cycling and bioaccumulation modeling have received attention in the past, with a strong focus on $MMHg^+$ bioaccumulation modeling. A model using the Shear realistic water column Turbulence Resuspension Mesocosms in the Beaufort Sea was presented by Kim et al. (2008); they incorporated the bioaccumulation and trophic transfer of $MMHg^+$ in a pelagic food web and the bentho-pelagic coupling to study the effect of sediment resuspension on $MMHg^+$ bioaccumulation. Schartup

et al. (2018) made a non-spatial model of MMHg$^+$ uptake and trophic transfer; they modeled bioaccumulation and trophic transfer of MMHg$^+$ at the base of the food web. Their non-spatial model is driven by observations of aquatic MMHg$^+$ and successfully reproduces observed bioaccumulated MMHg$^+$ concentrations in mesozooplankton. Zhang et al. (2020) developed a global model for MMHg$^+$ uptake in plankton; they modeled bioaccumulation by assuming an instant equilibrium between marine MMHg$^+$ and phytoplankton, and consequently modeled bioaccumulation into two functional groups of primary consumer zooplankton that were differentiated by their size. Bioaccumulation in zooplankton was estimated based on the concentration of MMHg$^+$ in the phytoplankton they consume and a size-specific MMHg$^+$ elimination rate. Rosati et al. (2022) published a model for the cycling and bioaccumulation of MMHg$^+$ in plankton in the Mediterranean Sea; they developed a coupled 3D Hg biogeochemical transport model to assess the cycling of Hg$^{2+}$, Hg$^0$, MMHg$^+$ and DMHg. Bioaccumulation was incorporated by modeling the bioconcentration of MMHg$^+$ in phytoplankton and the consequent biomagnification to zooplankton when they consume phytoplankton. Together, these models create a strong base for understanding the coupling of MMHg$^+$ in the marine environment to bioaccumulation at the base of the food web. This is expanded on by Bieser et al. (2023); which couples atmospheric Hg concentrations and deposition with MMHg$^+$ bioaccumulation in the mid-trophic level fish Herring (*Clupea harengus*), while taking into account the bioconcentration of Hg$^{2+}$ and MMHg$^+$ for all biota. This is done by integrating the MERCY v2.0 model to the output of the 3D ECOSMO-HAMSOM coupled system, which models biogeochemistry and hydrodynamics in the North and Baltic Seas on a spatial and temporal scale. In this way, the bioaccumulation of MMHg$^+$ could be coupled to atmospheric Hg cycling without having the models interact at run-time.

We implemented the bioaccumulation of Hg$^{2+}$ and MMHg$^+$ in a fully coupled model, which includes 2 trophic levels of fish. In addition to being fully coupled, it expands on previous models by including high-trophic-level fish while explicitly tracking the trophic level of all biota. Similarly to Bieser et al. (2023), we incorporate Hg cycling and bioconcentration and biomagnification of both Hg$^{2+}$ and MMHg$^+$ at every trophic level. We analyze bioaccumulation in the North and Baltic Seas in idealized 1D water columns, as this region provides varied hydrodynamical environments that we can use to test our model. Then we evaluate the importance of our findings in the same offline coupled 3D ECOSMO-HAMSOM-MERCY system presented in Bieser et al. (2023) to estimate the effect of bioaccumulation on Hg cycling and the Hg and MeHg budget.

The North and Baltic Seas are shelf seas in North-Western Europe. The Baltic Sea is a brackish sea of approximately 377,000 km$^2$, which is connected to the 575,000 km$^2$ large North Sea via the Danish straits. Both seas are important sources of seafood, and 2 million metric tonnes are landed annually (ICES, 2022). Hg input into the North and Baltic Seas is dominated by riverine input and atmospheric deposition (Kwasigroch et al., 2021).

In this study, we hypothesize that the ecosystem can influence Hg cycling in several ways, and our aim is to quantify the effect of the ecosystem on Hg cycling. To support this goal, we quantify the feedback of the ecosystem on marine Hg cycling by modeling a fully coupled Hg speciation and bioaccumulation model with and without bioaccumulation, the complexation of Hg with detritus and labile DOM, and the biogenic reduction of Hg$^{2+}$ to Hg$^0$ facilitated by cyanobacteria. Then we analyze the difference between the scenarios. In this article, we present the results of model runs in the North and Baltic Seas, using three idealized 1D water column setups that represent permanently mixed, seasonally mixed, and permanently stratified water column conditions. Additionally, we run the model in a 3D configuration of the North and Baltic Seas to analyze the spatial

**Table 1.** Definitions of Hg abbreviations.

| Abbreviation | Meaning |
|---|---|
| Hg | Refers to Hg in general |
| $Hg^{2+}$ | Dissolved Hg (Bioaccumulates) |
| $Hg^0$ | Elemental Hg (Volatile) |
| $MMHg^+$ | Monomethylmercury (Bioaccumulates, extremely toxic) |
| DMHg | Dimethylmercury (Volatile, extremely toxic) |
| MeHg | $MMHg^+$ + DMHg |
| tHg | All Hg, including what is bioaccumulated |
| tMeHg | All MeHg, including what is bioaccumulated |
| Bioaccumulated $Hg^+$ | All $Hg^{2+}$ that is bioaccumulation, does not include what is partitioned to detritus and DOM |
| Bioaccumulated $MMHg^+$ | All $Hg^{2+}$ that is bioaccumulation, does not include what is partMMitioned to detritus and DOM |
| Aquatic Hg | All Hg excluding what is bioaccumulated |
| Aquatic MeHg | All MeHg excluding what is bioaccumulated |

variation of the impact of the ecosystem on Hg cycling and quantify the overall effect of these interactions on the Hg budget of these seas. We investigated the role of the ecosystem on both the total Hg (tHg) and total MeHg (tMeHg) budget and the aquatic Hg and MeHg fractions. The tHg and tMeHg refer to all Hg and MeHg, including what is bioaccumulated. The aquatic Hg and aquatic MeHg refer to all Hg species that are in water but are not bioaccumulated. This is shown in Table 1 for clarity.

    The 3D configuration used in this study is based on the model of (Bieser et al., 2023), which is modified to investigate
scenarios in which we change the ecosystem drivers of Hg speciation to assess their impact.

## 2 Methodology

To evaluate the role of the ecosystem, we quantify the impact of several processes on total and aquatic Hg concentrations. In the first part of this section, we specify how these processes are implemented in the different scenarios that are simulated. The second part focuses on the models, setups, and parameterizations used in this study.

### 2.1 Processes

To quantify the impact of ecosystem interactions on marine Hg cycling, we evaluated three processes:

- Bioaccumulation

- Biogenic reduction

- Partitioning to detritus and labile-DOM

*Bioaccumulation* is the increase in $Hg^{2+}$ or $MMHg^+$ in the biota relative to the concentration in surrounding water. When Hg is bioaccumulated, it can no longer evaporate, undergo photolysis, or participate in chemical reactions that change its speciation; instead, it is transported with the organism that accumulated it. Thus, bioaccumulation removes aquatic $Hg^{2+}$ and $MMHg^+$, which can otherwise participate in chemical reactions that alter their speciation. The aquatic $Hg^{2+}$ and $MMHg^+$ are removed during the phytoplankton bloom and are released when the bloom period is over. This seasonal removal and release have the potential to influence Hg cycling.

*Biogenic reduction* is the reduction of $Hg^{2+}$ to volatile $Hg^0$ by cyanobacteria. It has been shown to be an important interaction during cyanobacterial blooms in the Baltic Sea (Kuss et al., 2015). Here, we attempt to quantify the impact that cyanobacterial blooms have on Hg cycling in the Baltic Sea. Biogenic reduction does not play a role in our North Sea setups, as there are no cyanobacteria in the North Sea in the ECOSMO E2E model (Daewel and Schrum, 2013). That agrees with the research of Kuss et al. (2015), where cyanobacteria-induced biogenic reduction was found in cyanobacteria that are specific to the Baltic Sea.

*Partitioning to organic carbon* is represented in the model by the binding of $Hg^{2+}$ and $MMHg^+$ to detritus and labile-DOM. Hg associated with organic carbon can be ingested by scavengers, contributing to the bioaccumulation process. Alternatively, when the detritus sinks, the Hg bound to it is transported to deeper water. In this way, the binding of Hg to organic carbon not only facilitates a flux of Hg to deeper water but also can deliver Hg from the upper water layers directly to scavenging animals. The only organic carbon particles for which the 1D setups account are detritus and labile-DOM originating from the ECOSMO E2E model.

The simulation scenarios of our research model are visualized in Fig. 1. Here we run the model with the following interactions enabled: (base case) without bioaccumulation but all other interactions (scenario A), without bioaccumulation and biogenic reduction (scenario B), without partitioning to detritus and labile-DOM (scenario C), and without any of the previously described interactions (scenario D). Since there is no biogenic reduction in the North Sea, the base case is only compared to scenarios B and C in the North Sea setups.

## 2.2 Models

In this study, we used two model systems with different purposes. The first system is idealized 1D water column setups, which
are used to generalize our findings. The second is a 3D setup, which is used to analyze the spatial patterns and estimate the
budgets of the North and Baltic Sea.

### 2.2.1 1D water column model

For the 1D system, the model design is shown in Fig. 1. We use the Generalized Ocean Turbulence Model (GOTM) to simulate
the hydrodynamics of the 1D water column setups, the ECOSMO E2E ecosystem model to simulate the marine ecosystem,
and the MERCY v2.0 model to simulate Hg cycling. (Daewel et al., 2019; Bieser et al., 2023; Burchard et al., 1999). The
models are coupled using the Framework for Aquatic Biogeochemical Modeling (FABM) (Bruggeman and Bolding, 2014).
ECOSMO E2E, MERCY v2.0, and bioaccumulation models are implemented through FABM and coupled to GOTM using
this framework. This coupling allows us to simulate the effect of the marine ecosystem from the ECOSMO E2E model on the
Hg cycling in the MERCY v2.0 model under the influence of the hydrodynamics from the GOTM model.

### 2.2.2 GOTM

The GOTM model is a 1D model that calculates the turbulence of a vertical 1D water column setup by computing the solutions
to the one-dimensional version of the transport equation of momentum, salinity, and temperature, while being nudged to
observational datasets. Therefore, our model has a vertical, but no horizontal resolution. GOTM calculates variables of the
physical state (temperature, salinity, and density), vertical transport (advection, diffusion, turbulence, and sinking) and surface
processes (surface elevation, friction, and velocity). GOTM communicates these state variables to the biochemical models
using the FABM interface.

### 2.2.3 MERCY

The MERCY v2.0 model is an intermediately complex Hg cycling model. It includes the speciation of $Hg^0$, $Hg^{2+}$, HgS,
$MMHg^+$, and DMHg in water, sediment, and biota. With this model, we implement the partitioning of $Hg^{2+}$ and $MMHg^+$ to
190 organic carbon and its speciation between the dissolved, particulate, and colloidal phases. In addition, atmospheric deposition
of Hg, its air-sea and sediment-water exchanges are also resolved in this model. In the 1D setups, the MERCY v2.0 model is
fully coupled with the ECOSMO E2E-GOTM system. In the MERCY v2.0 model, several drivers are incorporated to model the
spatial temporal variability of Hg speciation. All Hg species are treated as tracer variables and thus move with the movement
of water. Light is used to estimate the photolytic reduction rate ($Hg^0$ + photon $\rightarrow Hg^{2+}$), the photolytic oxidation rate ($Hg^{2+}$
+ photon $\rightarrow Hg^0$), and photolytic demethylation ($MMHg^+$ + photon $\rightarrow Hg^{2+}$, DMHg + photon $\rightarrow Hg^{2+}$, and DMHg + photon
$\rightarrow MMHg^+$). Temperature is used to estimate the temperature-dependent dark reduction of $Hg^{2+}$ ($Hg^{2+} \rightarrow Hg^0$). Furthermore,
the air-sea exchange of Hg in the MERCY v2.0 model is based on the approach used in Kuss (2014); Kuss et al. (2009), which

**Figure 1.** a) Schematic of the model setup. The black lines indicate the 1D setup where GOTM drives the ECOSMO E2E Ecosystem model and MERCY v2.0 Hg speciation model. These are used to simulate a base case and four scenarios with varying Hg–ecosystem interactions. The impact of the ecosystem is evaluated by comparing the base case to a scenario without: bioaccumulation (Scenario A), bioaccumulation and biogenic reduction (Scenario B), bioaccumulation and partitioning to detritus and DOM (Scenario C), and all mentioned ecosystem interactions (Scenario D).The purple lines show the 3D setup, where the HAMSOM model drives ECOSMO E2E and MERCY v2.0 models. The base case, Scenario A, and Scenario B are simulated in the 3D setup. b) Global map with the regional domain highlighted. c) Regional map of the North and Baltic Sea region. The 3D HAMSOM-ECOSMO-Mercy model domain is marked in blue. The three 1D setups, Northern North Sea (NNS), Southern North Sea (SNS), and Gotland Deep (GD), are labeled and marked with red points.

uses the temperature and salinity dependent Henry's law to estimate the equilibrium between atmospheric and marine $Hg^0$ concentrations and a wind-speed-dependent transfer rate.

### 2.2.4 The ECOSMO E2E ecosystem model

The ECOSMO E2E (ECOSystem Model End-to-End) ecosystem model (Daewel et al., 2019) is an extended version of the ECOSMO II model. ECOSMO E2E includes higher trophic levels, such as macrobenthos and fish. In the 3D setups, this model includes the following 7 biological functional groups: cyanobacteria, flagellates, diatoms, microzooplankton, mesozooplankton, fish, and macrobenthos. In the 1D setup, there are two functional groups of fish, 1 representing lower trophic level pelagic fish such as herring or sprat, and 1 representing a benthic top predator, such as cod. The model simulates nutrient cycles (nitrogen, phosphorus, and silicon), oxygen dynamics, and sedimentation processes. The ecosystem model and the MERCY model

interact in multiple ways. First, light absorption by phytoplankton detritus decreases available light, affecting light-dependent Hg speciation. Ecosystem variables are treated as tracers and move with water flow. Thus, if water currents carry biological variables, they also transport bioaccumulated Hg. Detritus is the only biological component with intrinsic movement, sinking at 5 m d$^{-1}$, while mesozooplankton and fish lack intrinsic movement. The ECOSMO E2E and the MERCY v2.0 model interact in multiple ways. First, light absorption by phytoplankton, detritus, and DOM decreases available light, affecting light-dependent Hg speciation and photosynthesis in deeper water. Ecosystem variables are treated as tracers and move with water flow, with the exception of fish 1 and fish 2, which both have no movement. Thus, if water currents carry biological variables, they also transport bioaccumulated Hg. Detritus is the only biological component with intrinsic movement, sinking at 5 m d$^{-1}$.

## 2.3 Modeled regions

To generalize our findings, the scenarios are tested using the 1D GOTM model in 3 setups with hydrodynamics common for coastal areas. They are designed around physical and biogeochemical regimes representing specific locations in the North and Baltic Seas. For this, we used a 1D ocean water column model to simulate a permanently mixed, a seasonally stratified, and a permanently stratified water column. This allows us to compare our findings across different physical and biogeochemical regimes.

### 2.3.1 Permanently mixed - Southern North Sea

The first North Sea setup is located in the Southern North Sea ($54°15'00.0"N\ 3°34'12.0"E$) with a depth of 41.5 m. This area is characterized by strong tidal mixing, resulting in high dissolved oxygen concentrations and temperatures throughout the water column. Remineralized nutrients are quickly available for phytoplankton production as they can be mixed with surface water throughout the year. The constant mixing of the water column results in the temperature fluctuating between 5 and 15 $°C$ for the whole water column depending on the season (Van Leeuwen et al., 2015). The setup location is 170 km away from the mainland, so despite being in the Southern North Sea, it does have characteristics of the open North Sea. This region is characterized by high primary production between 60 and 80 g C m$^{-2}$ y$^{-1}$ (Daewel and Schrum, 2013). The constant mixing allows macrobenthos to feed directly from the phytoplankton bloom, leading to a high macrobenthos stock (Heip et al., 1992). 41.5 m is also deep enough to support larger fish, such as herring and cod (Holm et al., 2022).

The available concentrations of dissolved silicate limit the growth of diatoms in the North Sea. Diatoms can dominate at the start of the bloom, but other phytoplankton taxa will take over the bloom after the silicate is depleted. The second flagellate bloom in the North Sea typically exceeds the initial bloom and can continue until light, nitrogen, or phosphorus becomes limiting (Peeters et al., 1991).

### 2.3.2 Seasonally mixed - Northern North Sea

The second setup is in the Northern North Sea ($57°42'00.0"N \ 2°42'00.0"E$) with a depth of 110 m. This setup has seasonal stratification in summer and complete mixing of the water column in winter (Van Leeuwen et al., 2015). This results in good growth conditions for phytoplankton in spring on the surface and a subsurface bloom in summer, when nutrients become depleted on the surface. In fall and winter, nutrients from the deeper water layers are mixed back into the surface. During summer, the temperatures in the Northern North Sea setup increase at the surface to 15-20 $°C$ but remain at 6 $°C$ in deeper water.

Primary production in the Northern North Sea is typically lower than in the Southern North Sea and ranges from 40 to 60 g C m$^{-2}$ y$^{-1}$ (Daewel and Schrum, 2013). Furthermore, seasonal stratification means that macrobenthos cannot feed directly from the phytoplankton bloom. This leads to smaller macrobenthos communities than in the Southern North Sea. Fish are better adapted to the Northern North Sea environment, which results in high stocks of fish. The Northern North Sea has a similar nutrient regime as the Southern North Sea, in which silicate limits diatom growth, and flagellates can be limited by light, nitrogen, or phosphorus (Peeters et al., 1991).

### 2.3.3 Permanently stratified - Gotland Deep

The third setup is the Gotland Deep ($57°18'00.0"N \ 20°00'00.0"E$) located in the central Baltic Sea. It is 249 m deep and is characterized by a permanent halo and oxycline. During spring and summer, an additional thermal stratification is established. This causes temperatures of 6 $°C$ and preserves anoxia in deep water. The surface water of the Gotland Deep setup remains oxic throughout the year and has temperatures fluctuating between 5 and 20 $°C$. The Gotland Deep setup is also considerably less saline than the North Sea and has a salinity of 12 and 7 g Kg$^{-1}$ in the deep and surface water, respectively (Nausch et al., 2003). This is caused by major freshwater riverine input combined with sporadic and limited inflow from the North Sea (Lehmann et al., 2021). Phytoplankton growth in the Baltic Sea is typically dominated by diatoms, but flagellates can form a large portion of the biomass in open-water areas, such as the Gotland Deep, while there can be an autumn bloom dominated by cyanobacteria. Primary production is estimated to be between 20 and 40 g C m$^{-2}$ y$^{-1}$ in this region (Daewel and Schrum, 2013). The open Baltic Sea is a productive fishing region with a fish biomass density similar to the North Sea; however, due to the anoxic bottom water, macrobenthos in the Gotland Deep is extremely limited. Studies in the Gdansk Deep found no macrobenthos with a population density significant for benthopelagic coupling under anoxic conditions, and similar conditions can be expected in the Gotland Deep (Kendzierska and Janas, 2024).

The Gotland Deep has silicate concentrations similar to those of the North Sea but with less nitrogen and phosphorus. Because of this, the available silicate is enough to support diatoms as the most abundant phytoplankton group in the Baltic Sea. The heavily stratified nitrogen-limited surface layer forms an ideal growth environment for cyanobacteria, which can fix nitrogen from the atmosphere. In this way, they can use the available phosphorus, until either phosphorus becomes limiting or the

270 end of the bloom season causes light to become limiting (Savage et al., 2010).

### 2.3.4 3D North and Baltic Seas

In addition to the idealized 1D setups of the different regimes, the total effect of the ecosystem on Hg cycling in the North and
275 Baltic Seas is quantified using a 3D model of the entire region. Both 1D and 3D models are used because the 1D models allow
us to analyze different ecosystem interactions in idealized circumstances, while the 3D models allow us to evaluate the total
effect of the ecosystem on Hg cycling, sedimentation, and evaporation in the model domain while assessing the effect of the
ecosystem on the total Hg budget of the North and Baltic Seas.

## 2.4 3D spatial model

In addition to 1D simulations, the effect of the ecosystem is analyzed by running the MERCY v2.0 3D Hg bioaccumulation
and speciation model described in Bieser et al. (2023) in the North and Baltic Seas with and without bioaccumulation and
biogenic reduction. MERCY v2.0 model uses the 3D hydrodynamic model HAMSOM (Hamburg Shelf Ocean Model) as a
physical model. In this setup, the HAMSOM-ECOSMO hourly output is used to drive the MERCY v2.0 model, making it
effectively an offline coupled system. The model is run from the beginning of 2011 to the end of 2015. The 3D HAMSOM-
285 ECOSMO-MERCY domain covers the Baltic Sea and the North Sea with open boundaries at the English Channel and at 63°N,
where the North Sea is connected to the Atlantic Ocean, as shown in Fig. 1. The resolution of the model is about 10 x 10
km$^2$ on a spherical grid with a vertical resolution of 20 layers. The upper four layers are 5 m thick, while the deepest layer
reaches a thickness of up to 250 m. The maximum water depth is 630 m. The first 4 years are used as spin-up and the final
year is used for the analyses. The model is run in its default setup, without bioaccumulation or biogenic reduction. The effect
of bioaccumulation on both tHg and tMeHg is visualized by plotting the relative difference in tHg and tMeHg caused by the
ecosystem. The data is visualized using the cartopy package in Python version 3.11.2.0.

## 2.5 HAMSOM

The HAMSOM (Hamburg Shelf Ocean Model) is a physical hydrodynamic ice-ocean model (Backhaus, 1983; Schrum and
Backhaus, 1999). It is directly coupled to the ECOSMO (ECOSystem MOdel) to form HAMSOM-ECOSMO (Schrum et al.,
2006). This coupling allows the ecosystem to directly influence the physical system, for example, through light absorption
by the biota. The HAMSOM-ECOSMO system is used to model the North and Baltic Seas. It is a comprehensive simulation
of both physical and biogeochemical drivers in the North and Baltic Seas and simulates key drivers such as riverine and
atmospheric nutrient input, horizontal advection and sedimentation, and burial of organic matter. This system provides the
physical, chemical, and ecosystem data that drive the MERCY v2.0 model.

 **2.6  Model development**

**Bioconcentration**

All biota in our model take up $Hg^{2+}$ and $MMHg^+$ from the water column due to bioconcentration. The bioconcentration rate for phytoplankton depends on the cell surface area and the diffusivity rate of $Hg^{2+}$ and $MMHg^+$ through the cell membrane (Mason et al., 1996). The surface area is estimated from the most common phytoplankton taxa in the three phytoplankton functional 305 groups for the North and Baltic Seas. The model organism and dimensions are shown in Table A1. The dimensions and shapes of phytoplankton are based on Olenina et al. (2003). The carbon content per cell was estimated from the calculated cell volume for diatoms as $pgC = 0.288\mu l^{0.811}$ and for flagellates and cyanobacteria as $pgC = 0.216\mu l^{0.939}$ (Menden-Deuer et al., 2000). Due to limited information on phytoplankton release rates, they are estimated based on uptake rates and observed concentrations (($Hg^{Aq}$ * uptake rate)/ $Hg^{Observed}$ = release rate) of $Hg^{2+}$ and $MMHg^+$.

Based on this, we implemented the change of bioconcentrated pollutant per day for a functional group ($Hg^{2+}$ or $MMHg^+$) via the following equation:

$$\frac{dC_{(p,g)}^{BC}}{dt} = b_g * C_p * r_{bc(g,p)} - (r_{rel(p,g)} + r_{bl(g)} + \sum_{z=1}^{n_z} r_{pred(z,g)}) * b_z * \frac{C_{(p,g)}^{BC}}{b_g} \tag{1}$$

$C_{(p,g)}^{BC}$ – concentration of pollutant p that is in the functional group g originating from bioconcentration [ng Hg m$^{-3}$];

$b_g$ – the biomass concentration of the functional group g [mgC m$^{-3}$];

$C_p$ – concentration of pollutant p [ng Hg m$^{-3}$];

$r_{bc(g,p)}$ – bioconcentration rate of group g of pollutant p [m$^3$ mgC$^{-1}$ d$^{-1}$];

$r_{rel(p,g)}$ – release rate of pollutant p by the functional group g [d$^{-1}$];

$r_{bl(g)}$ – biological loss rate due to mortality and respiration of group g [d$^{-1}$];

$n_z$ – total number of consumers that feed on group g;

$z$ – consumers that feed on group g;

$b_z$ - biomass concentration of consumer group z [mgC m$^{-3}$];

$r_{pred(z,g)}$ – the rate at which predator z feeds on group g [d$^{-1}$];

$t$ - time [day];

The bioconcentration, release, and turnover rates are shown in Table A3. In particular, Pickhardt and Fisher (2007) observes that $Hg^{2+}$ accumulates at similar levels in all phytoplankton, while the accumulation of $MMHg^+$ exhibits an inverse relationship with cell size. Due to this, the uptake/loss rate ratio is the same for all phytoplankton groups for $Hg^{2+}$; however, only the loss rate is the same for all phytoplankton groups for $MMHg^+$. The groups representing phytoplankton taxa with a smaller size and therefore a higher uptake rate also have a high $Hg^{2+}$ release rate. As a result, all phytoplankton groups reach equilibrium 330 at similar $Hg^{2+}$ concentrations. However, the equilibrium between uptake and release for $MMHg^+$ is inversely related to cell size. This means that smaller cells will accumulate more $MMHg^+$ per biomass while accumulating a similar amount of $Hg^{2+}$, compared to larger cells. The difference in the excretion rates of $Hg^{2+}$ and $MMHg^+$ can be explained by the different binding

behaviors of $Hg^{2+}$ and $MMHg^+$ in the cell. While $Hg^{2+}$ binds to smaller compounds and adheres to the cell wall, $MMHg^+$ binds to larger cytoplasmic proteins. This means that to excrete $MMHg^+$, it needs to be demethylated to $Hg^{2+}$, or excreted using dedicated peptide membrane channels (Bridges and Zalups, 2010; Takanezawa et al., 2023).

**Biomagnification in consumers**

The bioaccumulation in consumers depends on both bioconcentration and biomagnification. The release of bioconcentrated Hg is referred to as the release rate, and the release of biomagnified Hg is the turnover rate. This difference is caused by a different way of accumulation: bioconcentrated Hg can adsorb on the gills of fish where it can be released at different rates, compared to Hg absorbed in the gut tissues (Pickhardt and Fisher, 2007; Wang and Wong, 2003). In zooplankton, the same release and turnover rate is assumed. This is because zooplankton is so small that the bioconcentrated and biomagnified Hg can more easily homogenize throughout the animal (Tsui and Wang, 2004). The functional group-specific release and turnover rates are shown in Table A3.

The formulation for biomagnification is based on the consumption rates calculated by the ECOSMO E2E model, multiplied by an assimilation efficiency based on observations. The assimilation efficiency depends on the type of prey and is for $Hg^{2+}$ 0.2 when phytoplankton, 0.27 when consumers, and 0.13 when detritus or DOM is consumed. For $MMHg^+$, when phytoplankton, detritus, or DOM is consumed, assimilation efficiency is set for 0.80, and it is set to 0.96 when consumers are consumed (Mason et al., 1995; Tsui and Wang, 2004; Wang and Wong, 2003). The increase in Hg for consumer g per day is implemented into the model by the following equation:

$$\frac{dC_{(p,g)}^{BM}}{dt} = \sum_{s=1}^{n_s} (r_{pred(g,s)} * b_g * a_{(s,p)} * \frac{C_{(s,p)}}{b_s}) - (r_{to(p,g)} + r_{bl(g)} + \sum_{z=1}^{n_z} r_{pred(z,g)}) * C_{(p,g)}^{BM} \tag{2}$$

$n_s$ – total number of prey that group g feeds on;

$C_{(p,g)}^{BM}$ – concentration of pollutant p that is in the functional group g originating from biomagnification [ng Hg m$^{-3}$];

$r_{pred(g,s)}$ – consumption rate of group g on group s [d$^{-1}$];

$s$ – all functional groups that consume g predates on;

$a_{s,p}$ – assimilation efficiency of pollutant p when group s is consumed [unitless];

$C_{(s,p)}$ - the total concentration (from both biomagnification and bioconcentration) of pollutant p in group s [ng Hg m$^{-3}$];

$b_s$ - the biomass concentration of prey s [mgC m$^{-3}$];

$r_{to(p,g)}$ – turnover rate [d$^{-1}$];

Biomagnification follows the same loss process as bioconcentration, except that the turnover rate ($r_{to(p,g)}$) replaces the release rate ($r_{rel(p,g)}$)

Since this calculates the amount of bioaccumulation per volume, the bioaccumulation of pollutant p in group g per biomass ($B_{p,g}$) in ng Hg mgC$^{-1}$ can be calculated as:

$$B_{p,g} = \frac{C_{(p,g)}^{BM} + C_{(p,g)}^{BC}}{b_g} \tag{3}$$

It is important to note that extensive studies that separate the effects of bioconcentration and biomagnification are rare. Due to this, the estimated bioconcentration, turnover, and release rate of microzooplankton, mesozooplankton, and macrobenthos are all based on studies performed on pelagic water flea *Daphnia pulex*. Water fleas are abundant in the Baltic Sea, but not in the North Sea. Salinity does not appear to have a direct effect on the bioconcentration of Hg in zooplankton, and since Daphnia can have a size of 1-5 mm, it is average in size for zooplankton (Reinhart et al., 2018). The fish rates are based on a study investigating the uptake, release, and turnover rates in the Indo-Pacific fish Harry hotlips *Plectorhinchus gibbosus* (Wang and Wong, 2003).

### 2.6.1 A second fish functional group as top predator

One notable difference between the ECOSMO E2E model in Daewel et al. (2019) and used in (Bieser et al., 2023) and the version used in this study is the inclusion of a second fish functional group. The fish functional groups are referred to as fish 1 and fish 2. Both are parameterized by the same set of equations as described in the published ECOSMO E2E model, but differ in feeding preferences and rates (Daewel et al., 2019). Fish 1 preferably feeds on mesoplankton and microzooplankton and feeds on detritus and macrobenthos with lower preference. Fish 1 in the model represents a variety of smaller, mainly pelagic fish such as herring (*Clupea harengus*) and European sprat (*Sprattus sprattus*). The fish 2 group is modeled as a benthic top predator and prefers to eat macrobenthos, but can also feed on mesozooplankton, fish 1, and detritus. It is mainly representative of large demersal fish, such as cod (*Gadus spp.*), but would as a functional group also include members of other large benthic taxa such as whiting (*Micromesistius poutassou*) or haddock (*Melanogrammus aeglefinus*). Fish 2 is at the top of the food chain and is therefore not predated upon in the model. Instead, all loss terms are implicitly included in the mortality term. Table A2 shows the parameterization used in this study.

### 2.6.2 Trophic level modeling and parameterization

The ECOSMO E2E ecosystem model is originally designed to represent organic matter based on macronutrient fluxes, such as nitrogen, phosphate, and silicate fluxes. This means that certain interactions of the marine ecosystem that could biomagnify Hg, by increasing the trophic level of biota, such as predation of animals within the same functional group or even cannibalism, which do not alter nutrient fluxes or organic matter stocks, are not explicitly specified in the model (Montagnes and Fenton, 2012; Arrhenius and Hansson, 1996; Schrum et al., 2006). To quantify this, we explicitly resolved the trophic level of all biota. We modeled the trophic level in two different ways.

– **Trace organic carbon origin form biota** - Initially, we assumed that the trophic level of pelagic detritus and DOM is the same as that of the organisms from which it originates.

– **Standardize organic carbon trophic level as 1** - Afterward, we defined the trophic level of detritus and DOM as 1.0. This is done because Hg associated with detritus and DOM is assumed to have an instantaneous equilibration with Hg, rather than storing it as other state variables do.

Tracking the trophic level through the detritus and DOM provides the actual trophic level of biota, which can be compared with observations. The assignment of a fixed trophic level of 1.0 to the detritus and DOM emphasizes the number of trophic interactions that contribute to the biomagnification of Hg. Since Hg in sediment is tracked through organic carbon in our model, the trophic level of sediment detritus remains consistently modeled.

The modeled trophic levels are shown in Table 2. In the Gotland, Deep macrobenthos is absent because of the anoxic conditions. Except for fish 2 in the Northern North Sea, all functional groups have trophic levels that are lower than observed in the North and Baltic Seas. This was compensated for by reparameterizing the uptake efficiency of carbon used in the previous version of the ECOSMO E2E model. The uptake efficiency of carbon, known as assimilation efficiency, is a key parameter for biomagnification. Biomagnification occurs if the organic material is absorbed less efficiently or respired more efficiently than a pollutant, as this would result in an increase in this pollutant compared to organic material in the organism compared to its diet. The assimilation of carbon can be seen as two components; the first absorption refers to all carbon that is used by the fish and not directly excreted via faeces, whereas the assimilation refers to the carbon that is built up into the tissue of the fish. Fish are typically shown to have a 91-92% absorption efficiency, but only a 30-49% assimilation efficiency (Shelley and Johnson, 2022). Due to uncertainty, we parameterized the higher trophic-level fish with a lower assimilation efficiency than in the previously published ECOSMO E2E version, down to 45% in fish 2. To compensate for the decrease in carbon intake, the mortality and respiration rate of zooplankton is decreased to a value that is lower than in Daewel et al. (2019), but still within the range used in previously published models (Cruz et al., 2021). The phytoplankton is parameterized as shown in Table A1. To compensate for the increased zooplankton grazing, the growth rate is increased compared to the previously published ECOSMO E2E version, but remains within the experimentally observed range (Stelmakh and Kovrigina, 2021). All other values are the same as in (Daewel et al., 2019). This was done to tune the model to better reproduce higher MMHg$^+$ bioaccumulation, which is in line with observations. These interactions remain uncertain in the model, but replicating bioaccumulated concentrations is essential to estimate the bioaccumulation feedback on Hg speciation, which is the core focus of this study.

## 2.7 Model setup

### 1D setups

The 1D setups are run using the GOTM model mentioned earlier. The GOTM setups are based on observational data that is used to generate setups using iGOTM (https://igotm.bolding-bruggeman.com). All GOTM simulations run for the period 01-01-1989 12:00:00 until 01-01-2011 12:00:00, of which the first 10 years are considered a spin-up period and the period from 01-01-2000 12:00:00 to 01-01-2011 12:00:00 is the actual simulation period and is used for analyses. These setups are based on gridded bathymetry data for water depth (1/240° resolution) (GEBCO Bathymetric Compilation Group, 2020), ECMWF ERA5 data set for meteorological data (0.25°/hourly resolution) (Wouters et al., 2021), World Ocean Atlas for salinity and temperature profiles (0.25° resolution) (Garcia H.E. et al., 2019), and the TPOX-9 atlas for tides (1/30° resolution) (Egbert and Erofeeva, 2002). The setups have 1 grid cell m$^{-1}$, and the model is run using a forward Euler time-step ordinary differential

**Table 2.** Observed and modeled trophic level of functional groups. Observed values for microzooplankton, mesozooplankton, and fish 1 are based on Nfon et al. (2009), the trophic level for fish 2 is based on Jennings and Van Der Molen (2015), and for macrobenthos on Steger et al. (2019).

| Setup | Gotland Deep | | Northern North Sea | | Southern North Sea | | Observed |
|---|---|---|---|---|---|---|---|
| OC Trophic Level | modeled | 1 | modeled | 1 | modeled | 1 | - |
| Microzooplankton | 2.1 | 2.0 | 2.2 | 2.0 | 2.1 | 2.0 | 2.00 |
| Mesozooplankton | 2.2 | 2.2 | 2.8 | 2.5 | 2.6 | 2.5 | 2.87 |
| Fish 1 | 2.8 | 2.6 | 3.4 | 2.9 | 3.5 | 3.2 | 3.98 |
| Fish 2 | 3.7 | 3.5 | 4.2 | 3.7 | 3.8 | 3.5 | 4-4.2 |
| Macrobenthos | - | - | 2.8 | 2.3 | 2.6 | 2.3 | 2-4 |
| Detritus | 1.4 | 1.0 | 1.5 | 1.0 | 1.3 | 1.0 | - |
| DOM | 1.3 | 1.0 | 1.6 | 1.0 | 1.4 | 1.0 | - |
| Sediment | 1.4 | 1.3 | 1.4 | 1.3 | 1.4 | 1.3 | - |

equation that solves the state every 60 seconds. The variables are exported as daily mean values before being processed using
R v4.4.1. Plots are generated using ggplot v3.5.0. and the linear regression and statistics are calculated using ggpubr v0.6.0.

**Horizontal boundary exchange of Hg**

In GOTM, sediment and atmosphere are considered a horizontal surface that interacts with 1 cell of the surface grid. Therefore, we assume that burial will take place in 2 steps. The first step is sedimentation to the shallow sediment layer. From this layer, there is a fixed burial rate of 1.0E-5 $d^{-1}$ to deeper sediment that cannot be resuspended. These processes and rates are the
435 same for Hg, nutrients and organic carbon in the sediment. Sedimentation and resuspension of Hg are coupled with detritus in the ECOSMO E2E model. Hg bound to detritus will sink, sediment, and resuspend at the same rate as the organic matter it is associated with. Macrobenthos can also take up Hg bound to organic carbon when it consumes detritus or DOM. When macrobenthos lose Hg due to respiration or mortality, it is released into the sediment. Hg in the sediment has the same burial and resuspension rates as sediment carbon in the ECOSMO E2E model, as it is assumed to remain bound to organic carbon in
the sediment.

Due to its chemical properties, $Hg^0$ in the surface water layer is constantly exchanged with the atmosphere. This exchange is modeled the same as in Bieser et al. (2023) and modeled after Kuss (2014), which is based on the Henry's law constant determined by Andersson et al. (2008) to estimate the equilibrium between $Hg^0$ in the air and surface water. This method is extensively evaluated in Bieser et al. (2023) against measurements by Kuss et al. (2018). From all Hg speciations in aquatic
environments, two species are volatile $Hg^0$ and DMHg. The direction of atmospheric exchange depends on both the aquatic and atmospheric concentrations of those forms of Hg. Furthermore, the atmosphere can be a source of $Hg^0$ and $Hg^{2+}$, through direct wet deposition. To simulate the interactions of Hg with the atmosphere in this study, we used the approach previously used in the MERCY v2.0 model (Bieser et al., 2023). $Hg^0$ is constantly exchanged between the surface layer and the atmosphere.

When DMHg is present in the surface layer, it can evaporate from the marine system. In addition, there is the deposition of
$Hg^0$ and $Hg^{2+}$. Both atmospheric concentration and Hg deposition are provided by the CMAQ model, and the values are the same as in the 3D MERCY v2.0 (Bullock and Brehme, 2002; Bieser et al., 2023).

**Atmospheric deposition of nutrients**

Horizontal advection of nutrients plays an important role in the North and Baltic Seas. Nutrients are transported into the seas by rivers and are buried in sediment or transported to the Atlantic Ocean, and these processes are different for the three set-up
locations (Vermaat et al., 2008). Horizontal transport cannot be accurately captured in a 1D setup and is beyond the scope of this model. To mimic realistic wintertime nutrient concentrations of nitrate, phosphate, and silicate of 3.7, 0.3, and 5.0 $\mu$M in the Gotland Deep setup and 7.5, 0.5, and 5.0 $\mu$M in the North Sea setups, respectively (Burson et al., 2016). Both regions are characterized by strong horizontal influxes of nutrients that are not present in our 1D setups. To compensate for this limitation of our model, we chose atmospheric deposition so that it would create realistic nutrient conditions during winter in the surface
ocean throughout the simulation.

**Partitioning to DOM and detritus**

The only marine organic carbon particles in the water that interact with Hg in the 1D setups are detritus and DOM, which originate from the ECOSMO E2E model. The partitioning between Hg and detritus and DOM is assumed to be instantaneous, and equilibrium is forced on every model time step. This is based on $K_{dw}$ which is log(6.4) and log(6.6) for the partition of $Hg^{2+}$
into detritus and DOM responsibility and log(5.9) and log(6.0) for the binding of $MMHg^+$ to detritus and DOM responsibility. These values are based on Allison et al. (2005) and Tesán Onrubia et al. (2020) and evaluated in Bieser et al. (2023). This mechanism and rates are taken from the MERCY v2.0 model and are described in more detail in Bieser et al. (2023). Hg associated with organic carbon is taken up by organisms when they consume the detritus or DOM to which it is bound.

**Organic carbon to dry and wet weight conversion**

The biomass in our model is resolved in nutrients according to the Redfield ratio. In contrast, the uptake and release estimations are based on studies relying on weighing the samples and are therefore given in wet or dry weight. We assumed the ratio of milligram carbon to dry weight as: 0.2 for diatoms, 0.33 for flagellates and cyanobacteria, and 0.5 for zooplankton and fish (Walve and Larsson, 1999; Sicko-Goad et al., 1984). The weight values in this paper are always given in dry weight unless otherwise specified. Bioaccumulation can also be stated per wet weight in the literature (Nfon et al., 2009). The dry weight to
wet weight conversion factor of 0.2 is then used for phytoplankton, 0.3 for fish, and 0.16 for zooplankton and macrobenthos (Cushing, 1958; Ricciardi and Bourget, 1998; Cresson et al., 2017).

## 2.8 Initial conditions

The model is initialised with uniform conditions throughout the water. The model was initialised with low biomass of 0.1 mgC m$^{-1}$ for all phytoplankton, 0.01 mgC m$^{-1}$ for zooplankton, and 8.0E-5 for fish 1 and 8.0E-6 for fish 2. The model was spun up for 10 years to allow the model to stabilise. A notable difference in the initial conditions between the North Sea and Baltic Sea setups is the initial conditions for nutrients of 7.5 $\mu$M nitrate and 0.47 $\mu$M phosphate in the North Sea, compared to 5.0 $\mu$M nitrate and 0.31 $\mu$M phosphate in the Baltic Sea. The exact initial conditions used for all three setups are available via the YAML files that are provided and can be used to replicate the model output.

## 3 Model evaluation

Since this is the first publication to use specific 1D GOTM setups in combination with ECOSMO E2E and MERCY v2.0 coupled via FABM, and we made changes to the ECOSMO E2E model, we evaluated whether the 1D setups perform as expected and if the biological carbon and Hg species relevant for bioaccumulation are within the range of published 3D models. Data for higher trophic levels and bioaccumulation is much rarer and therefore this data is evaluated by a quantitative analysis if the modeled values are 1 standard deviation (SD) of observations. The 3D HAMSOM-ECOSMO-MERCY setup is extensively evaluated and shows good agreement with observations for Hg cycling as is discussed in more detail in Bieser et al. (2023), and therefore it was not further evaluated in this study.

### 3.1 1D Model and Observational Data

### 3.2 Evaluation of carbon fluxes

To evaluate carbon stocks and fluxes of the ECOSMO E2E model, we compared the modeled primary and secondary production to the production in the validated 3D ECOSMO E2E version in Daewel et al. (2019). In addition, we compared the model with observations for surface chlorophyll-a and zooplankton concentration, and the total fish and macrobenthos biomass. This comparison evaluated if our simplified 1D models remain consistent with a realistic ecosystem and is shown in Table 3.

The 3D ECOSMO E2E model estimates total primary production between 50 and 90 gC m$^{-2}$ y$^{-1}$ in the open North Sea and between 30-50 gC m$^{-2}$ y$^{-1}$ in the open Baltic Sea. The phytoplankton is initially driven by diatoms and succeeded by flagellates in the North Sea and a mix of diatoms, flagellates, and cyanobacteria in the Baltic Sea. Secondary production is estimated between 20-40 gC m$^{-2}$ y$^{-1}$ in the North Sea and 10-30 gC m$^{-2}$ y$^{-1}$ in the open Baltic Sea (Daewel et al., 2019).

The chlorophyll-a and biomass simulated in our 1D model are presented in Fig. 2. The total yearly primary production in our model is 50, 62, and 61 gC m$^{-2}$ y$^{-1}$, and the pelagic secondary production is 24, 42, and 29 gC m$^{-2}$ y$^{-1}$. This means that the primary and secondary production of the 1D model are in line with the previously published and validated 3D version of the model.

The average plankton concentration during the bloom is taken. The phytoplankton spring bloom period is selected as 1$^{st}$ of April - 30$^{th}$ of June and the zooplankton bloom period as 16$^{th}$ of April - 31$^{st}$ of October to select the majority of the bloom.

The average chlorophyll-a concentration and zooplankton biomass in the surface (0-10 m) are compared to observations. Chlorophyll-a levels in the Baltic Sea display significant variation. During bloom periods, the Northern Baltic Sea typically has values of 1-2 mg m$^{-3}$, whereas in the Southern Baltic Sea, values can reach 6 mg m$^{-3}$, with a basin-wide average of 2.64 mg m$^{-3}$ (OSPAR, 2017). During autumn, cyanobacteria can become the dominant taxa, but there is a large variety in the intensity of the bloom and the relative importance of different taxa (Hjerne et al., 2019). Our average modeled chlorophyll-a concentrations in the Baltic Sea of 0.92 mg m$^{-3}$ better resemble the Northern Baltic Sea than the Southern Baltic Sea. While the modeled chlorophyll concentrations in the North Sea are within the range of observations.

Zooplankton concentration ranges between 50 and 200 mgC m$^{-3}$ in the Northern North Sea and 0-50 mgC m$^{-3}$ in the Southern North Sea (Krause and Martens, 1990). Observations from the coast of Estonia, near the Gotland Deep, report 50 mgC m$^{-3}$ in measurements furthest from the coast (Ojaveer et al., 1998). The average concentration of zooplankton biomass during the bloom in our model falls within these ranges for all setups.

Total fish biomass in the North Sea is estimated to be between 15 and 23 g wet weight m$^{-2}$ for both the North Sea by Sparholt (1990) and Baltic Seas by Thurow (1997), or between 2.25 and 3.45 gC m$^{-2}$ assuming the earlier presented conversion rates of wet weight to carbon content of fish. This means that the modeled North Sea fish stocks are in agreement with observations, while the modeled fish population in the Baltic Sea is 7% higher than observed. The 7% can indicate the model overestimates fish in the Gotland Deep, but it is low enough that it can originate from uncertainty in the biomass estimate or be caused by uncertainty in the conversion from wet weight to carbon.

The peak and mean macrobenthos biomass is 12.3 and 6.84 gC m$^{-2}$ in the Southern North Sea and 3.4 and 0.99 gC m$^{-2}$ in the Northern North Sea, while macrobenthos biomass estimations range from 1.1 to 35.5 grams of carbon for the open North Sea, with the highest values closer to the coast (Daan and Mulder, 2001; Heip et al., 1992). So the macrobenthos biomass in our model aligns with observations. The Gotland Deep has anoxic deep water, so there is no macrobenthos in the Gotland Deep in our model, which matches observations (Kendzierska and Janas, 2024).

Overall the model produces biomass consistent with observations and the previously validated 3D version of the model. The only notable deviation is the Chlorophyll-a concentration of the Gotland Deep which closer resembles the Northern than Central Baltic Sea and the fish in the Baltic Sea is above the estimation made by Thurow (1997). This deviation, however, is only 7% which can also be caused by uncertainty in the original estimation or the conversion of the estimated wet weight to the modeled dry weight.

**Hg cycling; 1D Hg model results**

Table 4 shows the observed concentration of aquatic Hg and MeHg in the Baltic Sea measured by Kuss et al. (2017), while Fig. 3 shows total and aquatic Hg and MeHg in our 1D model. As mentioned in Table 1, tHg and tMeHg include bioaccumulated Hg and MeHg, while aquatic Hg and MeHg do not. The aquatic Hg includes Hg$^{2+}$, Hg$^{0}$, HgS, MMHg$^{+}$, DMHg and all Hg bound to detritus and DOM. The modeled tHg in the surface water of the Gotland Deep is 22% lower than the observations but will be within 1 standard deviation of the observed mean. For MeHg our modeled concentration is 50% below the observed mean

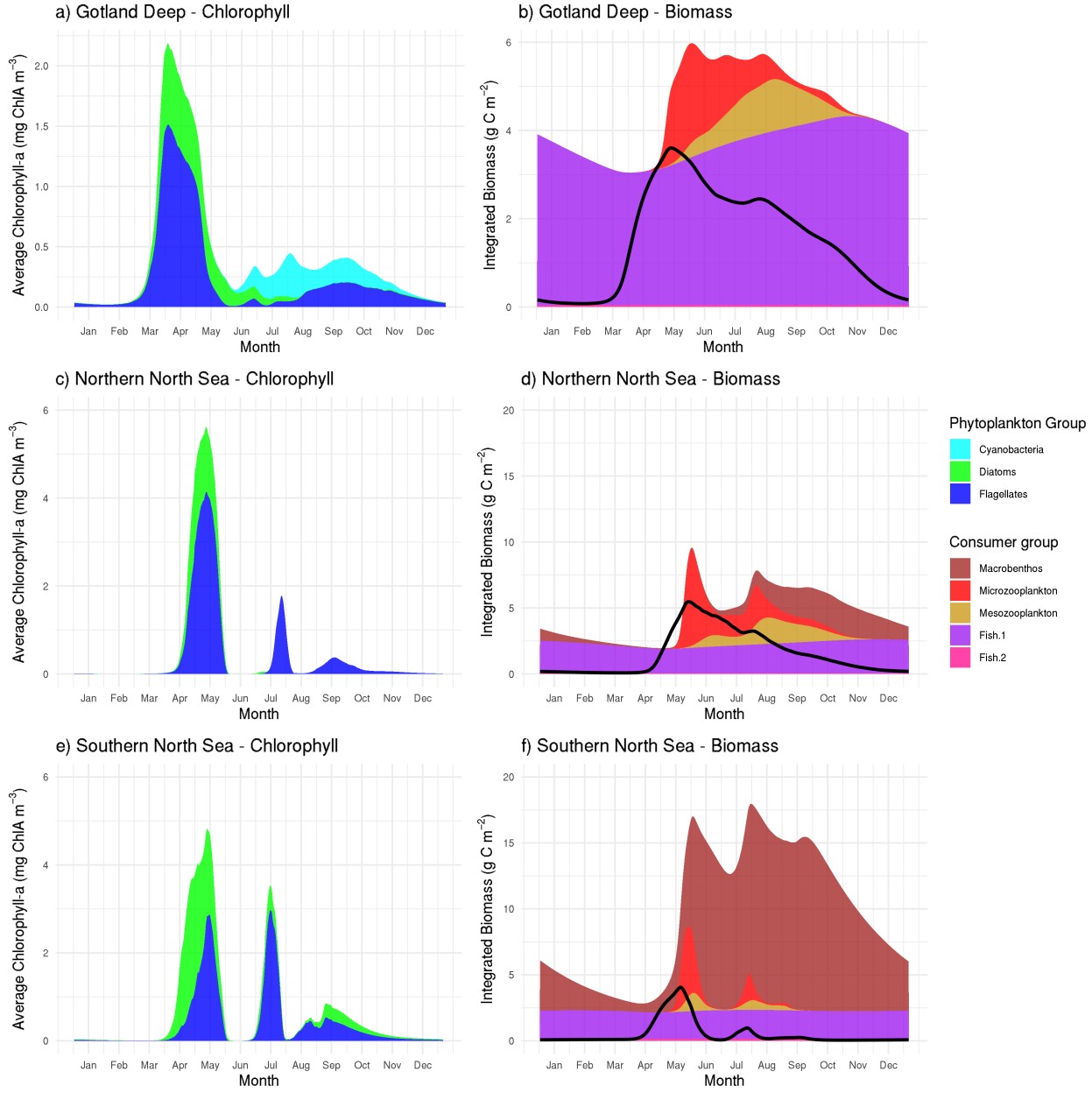

**Figure 2.** Daily means (Jan 2001–Jan 2011) of modeled chlorophyll-*a* (left panels) and organic matter (right panels). Living biomass is stacked; detritus and DOM are shown as a black line. Peak spring chlorophyll-*a*: (a) Gotland Deep, 2.2 mg m$^{-3}$ with a succesion from diatom to flagellate followed by cyanobacteria; (c) Northern North Sea, 5.6 mg m$^{-3}$ dominated by flagellates; (e) Southern North Sea, 4.8 mg m$^{-3}$ with diatoms early, followed by flagellates. At all sites, phytoplankton blooms are followed by increases in microzooplankton and then mesozooplankton. Fish biomass is stable: Northern North Sea (fish 1: 1.8–2.6 g C m$^{-2}$; fish 2: 0.044–0.054 g C m$^{-2}$), Southern North Sea (fish 1: 2.0–2.2 g C m$^{-2}$; fish 2: 0.14–0.16 g C m$^{-2}$), Gotland Deep (fish 1: 3.0–4.3 g C m$^{-2}$; fish 2: 0.40–0.44 g C m$^{-2}$). Macrobenthos varies seasonally: Northern North Sea 0.050–2.5 g C m$^{-2}$; Southern North Sea 0.64–13.8 g C m$^{-2}$; absent in Gotland Deep due to anoxia.

**Table 3.** Comparison of modeled and observed values for key ecosystem indicators across three regions. Observations of chlorophyll concentration are taken from OSPAR Commission (2017) while zooplankton biomass observations are based on Krause and Martens (1990) in the North Sea and Ojaveer et al. (1998)in the Baltic Sea. Estimations for for fish biomass are based on Sparholt (1990) in the North Sea and Thurow (1997) in the Baltic Sea.

| | Gotland Deep | | Northern North Sea | | Southern North Sea | |
|---|---|---|---|---|---|---|
| | Modeled | Observed | Modeled | Observed | Modeled | Observed |
| Surface Chlorophyll (mg m$^{-3}$) | $0.92 \pm 0.11$ | 1–6 | $1.78 \pm 0.12$ | 0.8–2.5 | $1.68 \pm 0.057$ | 0.8–2.5 |
| Surface Zooplankton biomass (mg m$^{-3}$) | $40.60 \pm 4.60$ | ~50 | $50.12 \pm 3.60$ | 50–200 | $34.63 \pm 5.40$ | 0–50 |
| Fish biomass (g m$^{-2}$) | $3.70 \pm 0.16$ | 2.25–3.45 | $2.29 \pm 0.17$ | 2.25–3.45 | $2.27 \pm 1.16$ | 2.25–3.45 |
| Macrobenthos biomass (g m$^{-2}$) | 0 | 0 | $0.99 \pm 0.33$ | 0.6–17.5 | $6.84 \pm 1.16$ | 0.6–17.5 |

and 17% below 1 standard deviation of the observed mean. This is hard to evaluate, as the measurement protocol does not lyse phytoplankton cells; we assume that no MeHg associated with biota is measured. However, if we assume that MeHg associated with phytoplankton is measured, our modeled concentration will be above the observed mean with $0.12 \pm 0.08$ pmol. Because of this uncertainty, we also evaluated the MeHg content indirectly through bioaccumulation. To this extent, Table 4 shows the modeled and observed concentration in diatoms and the observed volume concentration factor (VCF). Since the concentration of MeHg in diatoms in the Gotland Deep ($14.4 \pm 1.05$) is within 1 standard deviation (SD) of the observed mean ($10.0 \pm 5.0$), and the modeled VCF in Gotland Deep ($1.22E5 \pm 3.8E4$) is within the observed range (2.0E4 - 6.4E6), and the observed concentration of MeHg ($0.10 \pm 0.04$) is between our modeled concentration with ($0.12 \pm 0.08$) and without ($0.05 \pm 0.09$) MeHg in phytoplankton, we conclude that our model produces aquatic Hg and MeHg values that are in line with observations, with the caveat that the evaluation of the aquatic MeHg content is based on a limited amount of data with a high measurement uncertainty. Figure 3 shows the aquatic and total Hg and MeHg values modeled and demonstrates the difference between the total and aquatic Hg fraction in both depth and time. Notable is the small difference in the total and aquatic fractions of Hg compared to the large difference for MeHg in all three setups. In surface water during the bloom period, phytoplankton take up MeHg, which decreases aquatic MeHg. The removal of MeHg from the water prevents its (photo)demethylation and the dissolved MeHg fraction is replenished by the methylation of Hg$^{2+}$. This increases the tMeHg content, while this is not the case for tHg. An additional noteworthy observation is in the shallow well-mixed Southern North Sea. The mixing allows macrobenthos to feed directly off the spring bloom and thus removes both organic material and Hg from the water column. This leads to a drop in tMeHg during the bloom. In winter, the MMHg$^+$ bioaccumulated by macrobenthos is resuspended, leading to a very high wintertime concentration of both aquatic and total MeHg in winter in the Southern North Sea.

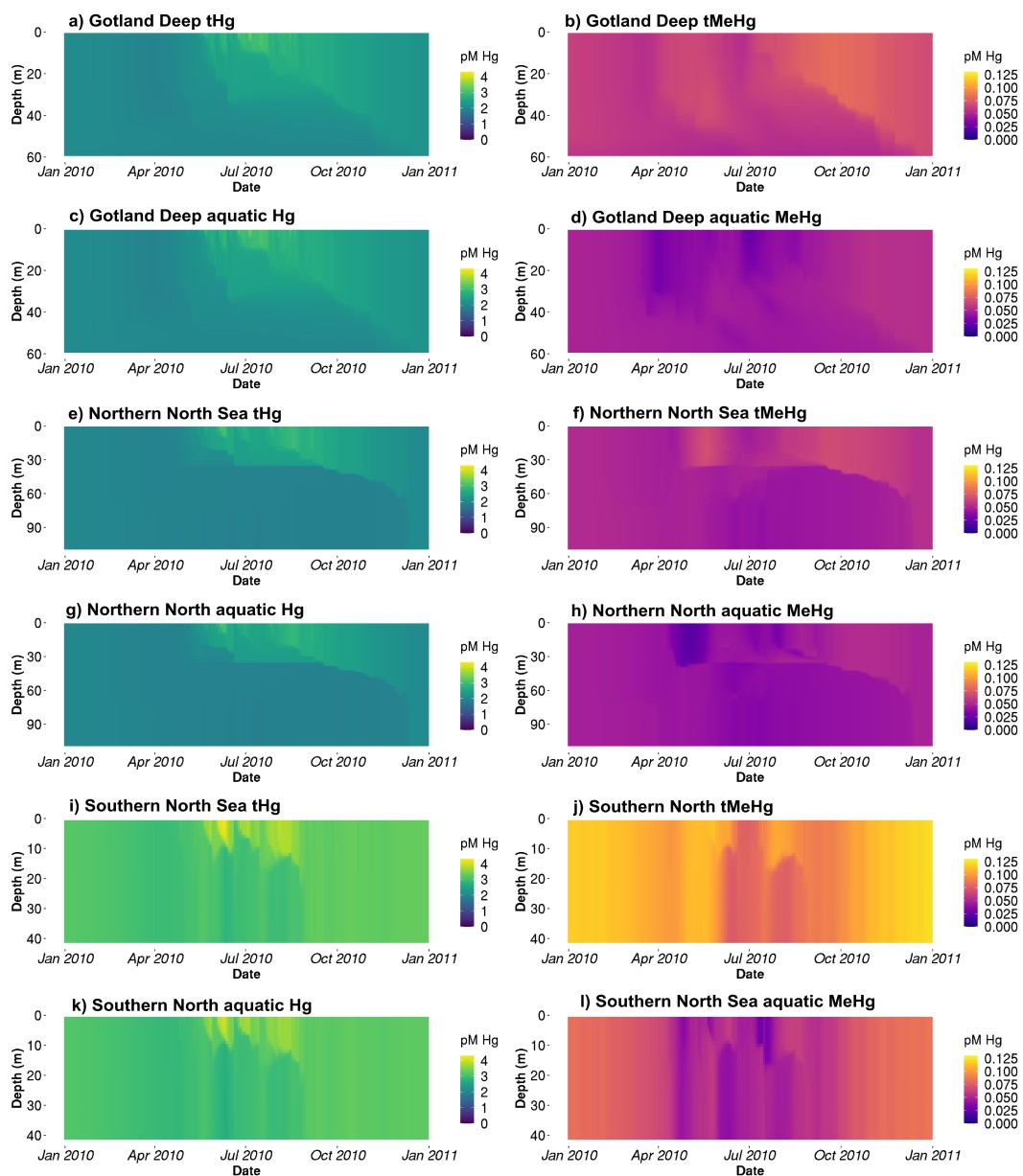

**Figure 3.** The tHg and tMeHg and the aquatic Hg (Hg$^{2+}$, Hg$^{0}$, HgS, MMHg$^{+}$, and DMHg) and MeHg (MMHg$^{+}$ and DMHg) concentrations in the last year of the simulation (Jan 2010-Jan2011) in all setups. The Gotland Deep and the Northern North Sea have a similar total Hg concentration of 2-3 pM, with an increase in Hg of up to 3.5 pM in the Northern North Sea. The concentration of Hg in the Southern North Sea is 2.5-3.5 pM during most of the year and reaches 4 pM during the spring bloom. The tMeHg content is 0.04-0.06 pM in the surface of the Gotland Deep and Northern North Sea and up to 0.08 pM in the Southern North Sea during winter. A clear decrease in all setups in aquatic MeHg during the bloom period can be seen, while the total concentration remains more stable.

## Bioaccumulation model results

The modeled Hg bioaccumulation for biota in the Gotland Deep was compared to observations, because the Baltic Sea is studied more extensively for Hg cycling, and studies such as performed by Kus2014Exchange provide the opportunity to validate Hg cycling, while studies such as Nfon2009TrophodynamicsSea allow the validation of low-trophic-level biota, while data on Hg cycling and bioaccumulation in low-trophic-level biota are extremely limited in the North Sea. For this evaluation, we only used model output where the biomass exceeded 100 mg carbon m$^{-2}$ and took the modeled values of the <5 m for phytoplankton and <20 m for other biota to ensure that our bioaccumulation values resemble the modeled values in surface water where biota are active. A comparison between our modeled and observed bioaccumulation is shown in Table 4. Additionally, year-round bioaccumulation of Hg$^{2+}$ and MMHg$^{+}$ in the Gotland Deep, the Northern North Sea, and the Southern North Sea is shown in the supplementary information in Figures S1, S2, and S3 respectively.

**Table 4.** Evaluation of the modeled bioaccumulation. Modeled and observed aquatic Hg and MeHg concentrations are shown first based on Kuss et al. (2017), then the observed range of the VCF in diatoms is shown based on Lee and Fisher (2016) and the mean and standard deviation of the modeled VCFs for MeHg into diatoms are shown. The observations for pelagic invertebrates are based on Nfon et al. (2009), the observations for fish on Kwaśniak et al. (2012) and Polak-Juszczak (2018) and the observations for macrobenthos on measurements of *Macoma Baltic* in the Baltic Sea (ICES 25 region) by Polak-Juszczak (2012). Of both model and observations the mean, standard deviation (SD) and coefficient of variation (CV) is shown. The concentration of aquatic tHg and MeHg are shown in pM, the VCF is unitless and the mean and SD of both bioaccumulated Hg$^{2+}$ and MMHg$^{+}$ is shown in ng g$^{-1}$ d.w. while the CV is unitless.

| Variable | Observed | Model; Gotland Deep | Model; Northern North Sea | Model; Southern North Sea |
|---|---|---|---|---|
| Aquatic Hg | 1.65 ± 2.1 | 2.37 ± 0.61 | 2.26 ± 0.22 | 2.97 ± 0.16 |
| Aquatic MeHg | 0.10 ± 0.04 | 0.05 ± 0.09 | 0.06 ± 0.01 | 0.04 ± 0.01 |
| VCF MeHg in diatoms | 2.0E4-6.4E6 | 1.22E5 ± 3.8E4 | 1.0E5 ± 4.58E4 | 1.1E5 ± 2.5E4 |
| Diatoms (tHg) | 10.0 ± 5.0 (0.50) | 14.4 ± 1.05 (0.07) | 12.9 ± 2.11 (0.16) | 16.1 ± 1.62 (0.10) |
| Microzooplankton (tHg) | 37.5 ± 31.3 (0.83) | 47.0 ± 12.1 (0.26) | 50.3 ± 10.8 (0.22) | 54.4 ± 19.7 (0.36) |
| Mesozooplankton (tHg) | 62.5 ± 12.5 (0.20) | 66.1 ± 13.5 (0.20) | 68.8 ± 10.8 (0.16) | 68.3 ± 17.7 (0.26) |
| Fish 1 (Hg$^{2+}$) | 12.7 ± 7.0 (0.55) | 7.17 ± 1.72 (0.24) | 7.95 ± 0.82 (0.10) | 11.8 ± 1.38 (0.12) |
| Fish 1 (MMHg$^{+}$) | 49.3 ± 30.7 (0.62) | 31.8 ± 1.90 (0.06) | 21.2 ± 0.92 (0.04) | 44.4 ± 3.62 (0.08) |
| Fish 2 (Hg$^{2+}$) | 34.3 ± 23.7 (0.69) | 4.20 ± 0.82 (0.20) | 4.81 ± 0.65 (0.14) | 10.0 ± 1.84 (0.18) |
| Fish 2 (MMHg$^{+}$) | 180 ± 72.3 (0.40) | 65.0 ± 2.17 (0.03) | 37.0 ± 0.95 (0.03) | 64.5 ± 3.43 (0.05) |
| Macrobenthos (tHg) | 25.0 ± 21.4 (0.84) | – | 20.0 ± 6.6 (0.22) | 30.1.5 ± 3.2 (0.16) |

## Evaluation of modeled bioaccumulation

Table 4 shows the modeled and observed means, the standard deviation (SD), the coefficient of variation (CV) and the VCF for MeHg in diatoms. The modeled bioaccumulation values from surface water (0-5 m) for diatoms and (0-20 m) for consumers

are used for the evaluation. Because there are high-quality observations for the Baltic Sea, but not for the North Sea, we focused the evaluation efforts on the Baltic Sea. The observations are for tHg in invertebrates, while we have separate observations of $Hg^{2+}$ and $MMHg^+$ in fish. Fish 1 in our model is compared to the observations of herring and fish 2 and to observations in Atlantic Cod. With the exception of the bioaccumulation into fish 2, all bioaccumulation values are within one observed SD of the observed mean. This, combined with the high CV of the observations, shows that our modeled values are well within the plausible range based on observations, except for the bioaccumulation in fish 2, which is lower than observed. It should be noted that the bioaccumulation modeled in fish 2 is not out of the observed range. Observations for tHg bioaccumulation in Atlantic Cod in the Baltic Sea range between 3 and 1567 ng $g^{-1}$ d.w. (ICES, 2020). However, the observed mean $MMHg^+$ content of all datasets is more than 1 observed SD higher than our modeled mean, leading us to believe that our model underestimates bioaccumulation in fish 2.

**Modeled biomagnification**

In Fig. 4 we plotted the correlation between the trophic level and the accumulated $Hg^{2+}$ and $MMHg^+$. To assess biomagnification in bioaccumulated $MMHg^+$, we assumed an exponential trend between the bioaccumulation of $MMHg^+$ and the trophic level, which is forced through the first consumer (microzooplankton). In this way, the exponent is the biomagnification factor. Observed BMF are between 2.0 and 3.4 in the North Sea for $MMHg^+$ (Baeyens et al., 2003). For $Hg^{2+}$ an exponential relationship between trophic level and bioaccumulation was neither expected nor present; therefore, we correlated this using a second degree polynomial. The modeled biomagnification factor is between 2.1 and 3.1 for $MMHg^+$. This means that our modeled biomagnification factor for $MMHg^+$ is within the range of observations in the North Sea.

**Summary of the model evaluation**

Based on the evaluation of the bioaccumulation, we conclude that the general trend of high bioaccumulation of $MMHg^+$ and low $Hg^+$ is well reproduced in our model. However, the performance of the model is lower in higher trophic levels and underestimates bioaccumulation into Atlantic Cod. The underestimation in modeled $MMHg^+$ values in cod can be attributed to the ecosystem model that underestimates the trophic level of cod by 0.5-0.7. If we extrapolate the equation for biomagnification in the Gotland Deep shown in Fig. 4 ($15.42e^{0.97*(TL-2)}$), in which TL is the trophic level, and estimate bioaccumulation at the observed trophic level Atlantic Cod (4.2), we would expect a $MMHg^+$ content of 123 ng Hg $g^{-1}$ d.w., which would be well within 1 observed SD of the observed mean. Because our model accurately predicts the tHg content of plankton, the content of $Hg^{2+}$ and $MMHg^+$ in fish 1, and the $MMHg^+$ content at high trophic levels according to their trophic level, it appears that the bioaccumulation model accurately models $MMHg^+$ bioaccumulation based on trophic interactions, but that our fish 2 better resembles a mid-level trophic predator both in trophic position and $MMHg^+$ than a high trophic level predator such as Atlantic Cod.

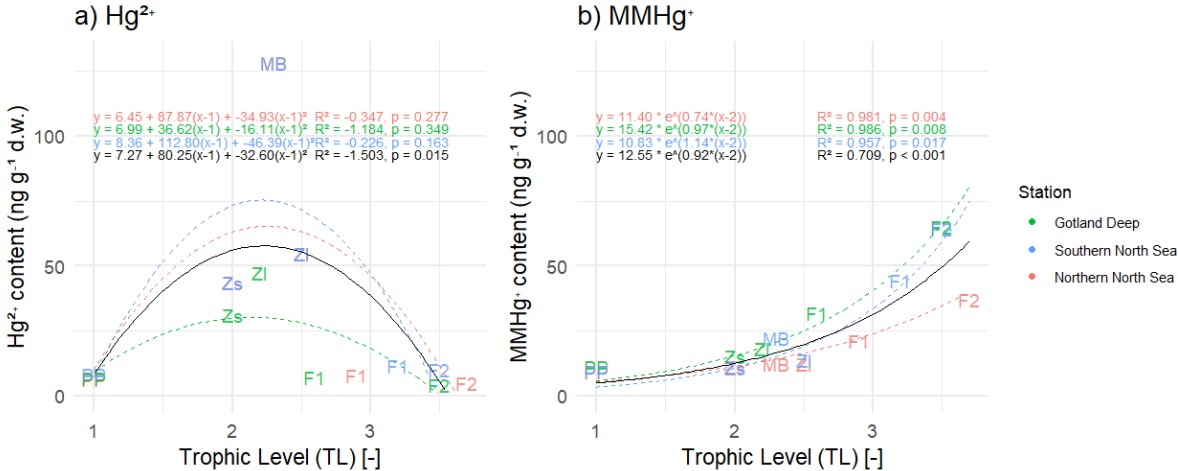

**Figure 4.** Figure 4 a) shows the correlation between trophic level and Hg$^{2+}$ and Fig. 4 b) shows the correlation between trophic level and bioaccumulation of MMHg$^+$ in all three setups, the black indicates the average. PP stands for primary producers, Zs for microzooplankton, Zl for mesozooplankton, F1 for fish 1, F2 for fish 2, and MB for macrobenthos. The correlation is fitted that is forced through microzooplankton. The R$^2$ of the correlation between trophic level and Hg$^{2+}$ shows that there is no correlation (R$^2$<0.00) in the Northern North Sea and Southern North Sea. In all setups, there is a strong correlation between trophic level and MMHg$^+$ bioaccumulation (R$^2$>0.95). Interestingly, the correlation between bioaccumulation MMHg$^+$ and trophic level for all setups averages is considerably lower (R$^2$=0.709 for MMHg$^+$) than for the setups separate.

## 4    Results & Discussion

### 4.1    The effect of bioaccumulation on tHg and tMeHg concentrations

The 10-year average difference in the daily mean tHg and tMeHg between the base case and the setups without bioaccumulation and biogenic reduction in the Gotland Deep is shown in Fig. 5. Additionally, Table 5 shows the mean difference of tHg and tMeHg, the mean and range of the bioaccumulated Hg fraction in the bioaccumulation setup, and the average annual amplitude of the difference in tHg and tMeHg, calculated as half the total range of values. This is shown because, while the average differences provide information on the net effect of bioaccumulation, the amplitude highlights seasonal variations. If the ecosystem causes a large average difference, but this difference is consistent throughout the year, the amplitude will be 0%, but if the ecosystem causes an increase of 100% and consequently a decrease of 100% with no average effect, the amplitude will be 100%.

**The effect of bioaccumulation on tHg**

Bioaccumulation has only a small effect on tHg with a maximum increase of 3% while the average percentage of tHg that is bioaccumulated is 1% in every setup. This increase is caused because the biota takes up Hg$^{2+}$, which is consequently protected

against reduction to $Hg^0$. Since $Hg^0$ in the water is in exchange with atmospheric $Hg^0$, a decrease in $Hg^0$ will result in a decrease in evaporation until the concentration of $Hg^0$ in the water is re-equilibrated with the atmospheric concentration. At this point, the atmospheric exchange and the aquatic Hg concentration in the scenarios with and without bioaccumulation are similar, while the tHg concentration is up to 3% higher in the scenario with bioaccumulation. This is further shown in Fig. 5. which shows the seasonal difference for both aquatic and total Hg and MeHg. Figure 5 shows that even though the difference is small, there is always more tHg in the base case than in scenario B without bioaccumulation. The only exception is in the permanently mixed shallow Southern North Sea in summer and autumn, which has a large amplitude in the difference in tHg with an increase of 11% at the beginning of the bloom period and a decrease of 6% later in the bloom. This decrease in tHg is caused by macrobenthos. which feeds directly on the plankton bloom and transports Hg from the water column to the benthic system via the consumption of organic material. During late autumn,winter, and early spring, this tHg is released back into the water resulting in the above-mentioned high increase in tHg of 11% at the onset of the phytoplankton bloom. This causes bioaccumulation to cause a similar mean increase of 3% in the Southern North Sea compared to the other setups, while the amplitude of the difference is 9% compared to 1% in other setups, demonstrating a strong seasonal effect.

**The effect of bioaccumulation on tMeHg**

tMeHg increases by 41, 23, and 46% in the Gotland Deep, Northern North Sea, and Southern North Sea respectively. This is very similar to the percentage of tMeHg that is bioaccumulated, which is 41, 29, and 43% respectively. Bioaccumulation increases tMeHg, as bioaccumulated $MMHg^+$ cannot be photodegraded, and aquatic MeHg is replenished by additional methylation. During autumn and winter, the biomass decreases and bioaccumulated $MMHg^+$ is released back into the water. During this period, the light intensity and therefore photodegradation are lower, and detritus and DOM concentrations are high, which leads to additional shading further reducing the photodegradation rate. This causes the aquatic tMeHg concentration to be higher outside of the bloom period. This means that an increase in bioaccumulated $MMHg^+$ equals a smaller decrease in aquatic MeHg, causing an increase in tMeHg. Additionally, bioaccumulation can increase aquatic MeHg by removing dissolved MeHg when photodegradation is high and re-releasing this MeHg when photodegradation is low, which explains the increase in aquatic tMeHg after the bloom period in the scenario with bioaccumulation compared to the scenario without bioaccumulation.

This increase in tMeHg is driven by organic material and shows a linear relationship with biomass. The correlation of the daily 10-year average values for the increase in tMeHg and tHg with biomass in the surface (0-20 m) is shown in Table 6. If we combine the three setups, we find an increase in tMeHg of 0.028 ng Hg $mgC^{-1}$ (r=0.47,p<0.001) or a relative increase of 1% per 4.5 mgC $m^{-3}$ (r=0.59, p<0.001) and an increase in tHg of 0.068 ng Hg $mgC^{-1}$ (r=0.07,p<0.001) or a relative increase of 1% tHg per 71 mgC $m^{-3}$ (r=0.07, p<0.001). Although both correlations are significant at the 99% confidence level, the relationship between the difference in tMeHg and biomass is notably stronger (r=0.47) than for the difference in tHg (r=0.07). Although bioaccumulation is a large amount of tMeHg, it does not constitute a large amount ($<$ 5%) of tHg; therefore, the formation of MeHg due to methylation is not decreased and the bioaccumulated MeHg is replenished due to its rapid equilibrium with other Hg species.

**Table 5.** The percentage difference in tHg, and tMeHg average concentrations caused by the ecosystem, and the average seasonal amplitude in surface water (0-20m). Additionally the mean and range of the percentage of tHg and tMeHg that is bioaccumulated (Hg(Bio) and MeHg(Bio)) in the scenario with bioaccumulation is shown. The percentage is calculated as the (scenario - base case)/((base case + scenario)/2)*100. Negative values indicate a decrease caused by the ecosystem effect, and positive values indicate an increase.

| Setup | tHg | tHg$_{Amp}$ | tHg(Bio) | tMeHg | tMeHg$_{Amp}$ | MeHg(Bio |
|---|---|---|---|---|---|---|
| Gotland Deep (no bioaccumulation) | 3% | 1% | 1(0-2)% | **41%** | 16% | 41 (26-45)% |
| Northern North Sea (no bioaccumulation) | 1% | 1% | 1(0-5)% | **23%** | 24% | 29 (13-59)% |
| Southern North Sea (no bioaccumulation) | 3% | 9% | 1(0-4)% | **46%** | 28% | 43 (31–64)% |
| Gotland Deep (no biogenic reduction) | -8% | 5% | 1(0-2)% | **29%** | 16% | 41 (26-45)% |

**Table 6.** Correlation, slope, and p-value for the difference in tMeHg and tHg vs the total biomass per setup. There is a much stronger correlation (r=0.547-0.809) between the increase in tMeHg then the increase in tHg (r=0.114-0.149). The slope for tMeHg ranges from 0.134 and 0.25 and for tHg between -0.0631 and 0.0279 mgC$^{-1}$ m$^3$

| | tMeHg | | | tHg | | |
|---|---|---|---|---|---|---|
| Setup | Correlation (r) | Slope | p-value | Correlation | Slope | p-value |
| Gotland Deep | 0.809 | 0.252 | <0.001 | 0.149 | -0.0631 | <0.001 |
| Northern North Sea | 0.739 | 0.134 | <0.001 | 0.218 | 0.00691 | <0.001 |
| Southern North Sea | 0.547 | 0.226 | <0.001 | 0.114 | 0.0279 | <0.001 |

Macrobenthos in the Southern North Sea have a similar, although stronger, effect on tMeHg concentrations to that of tHg. However, this effect is partially altered by the strong increase in tMeHg as a result of the high biomass in this setup. In spring in the Southern North Sea, there is a 77% increase in tMeHg due to bioaccumulation, which is the highest of all setups. However, this rapidly drops as the bloom progresses, and macrobenthos feed on the plankton bloom, causing the increase in tMeHg due to bioaccumulation to briefly be the lowest in the Southern North Sea at the end of the bloom. During winter, the macrobenthos population decreases and tMeHg is re-released, causing a large increase in wintertime tMeHg, making the increase in tMeHg due to bioaccumulation in the Southern North Sea the largest in all setups.

**The effect of bioaccumulation on the Hg budget and horizontal transport**

In Fig. 6, the difference in the content of tHg and tMeHg and the export by sedimentation and evaporation from the North and Baltic Seas caused by bioaccumulation is evaluated using the 3D the MERCY-HAMSOM-ECOSMO setup. There is a decrease in the evaporation of 9% (Fig. 6b) and a decrease in the burial of 19% (Fig. 6i) caused by bioaccumulation. Figures 6e and Fig. 6f show tMeHg and the difference caused by bioaccumulation. While there is an average increase caused by the bioaccumulation of 10% in tMeHg, this is not uniform. There is a decrease in the Baltic Sea, an increase in the North Sea, and a very strong

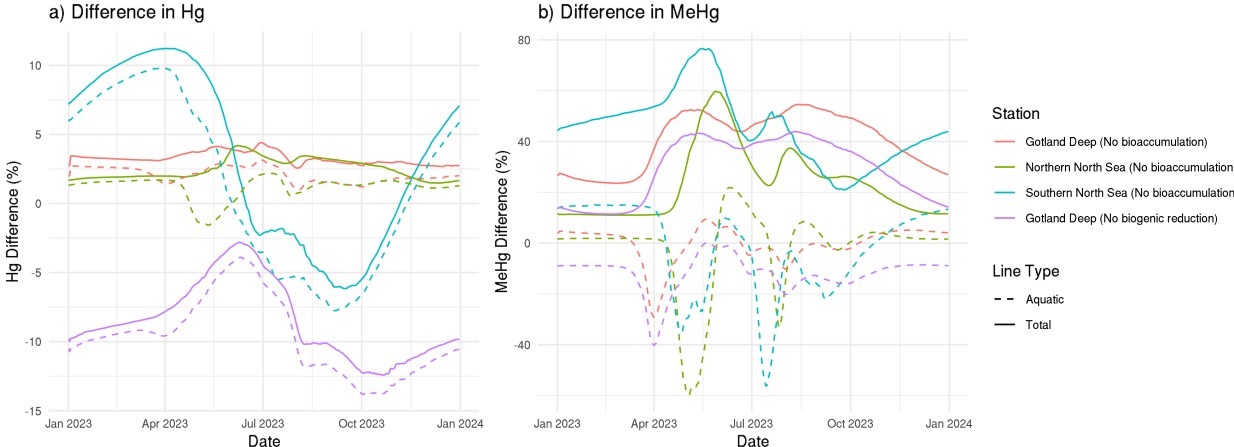

**Figure 5.** The difference in tHg, aquatic Hg, tMeHg, and aquatic MeHg in the base case and the scenario without bioaccumulation and the scenario without biogenic reduction in the Gotland Deep in the top surface 20 meters of the water column. For tHg there is a strong seasonal effect where it is decreased during summer and autumn and increased during spring. This is caused by the consumption of Hg-containing organic material during spring and summer and the release during winter. There is no notable effect ($> 5\%$) in the tHg in the Northern North Sea and Gotland Deep due to bioaccumulation. In the Gotland Deep, however, biogenic reduction causes a strong decrease of up to 12%. There is no large difference ($>5\%$) between the difference in aquatic Hg and tHg. The difference in tMeHg is much more pronounced. There is a consistent increase in tHg which peaks during spring and summer. These peaks in an increase in tMeHg coincide with a smaller decrease in aquatic MeHg.

increase in the Danish Straits. This, combined with the decrease of both evaporation and burial, indicates that bioaccumulation facilitates a flux of Hg out of the Baltic Sea toward the North Sea and the Atlantic Ocean. We see an average decrease of 2.3 and 57 pmol m$^{-2}$ y$^{-1}$ in evaporation and burial, respectively. Our model shows that the bioaccumulation into plankton keeps Hg pelagic, which facilitates transport to the North Sea and consequently the Atlantic Ocean. Without bioaccumulation, a fraction of this Hg will evaporate or be bound to POC and be buried. The decrease in tHg due to bioaccumulation is strongest in areas such as the Wadden Sea and the Bay of Bothnia, where there are both large riverine inputs of Hg and high primary production. However, this local effect is relatively small and we see a total increase in export of 14 kg Hg y$^{-1}$ to the Atlantic Ocean due to bioaccumulation. Of this 14 kg, 13 kg would be buried in sediment, and 1 kg would be evaporated into the atmosphere without bioaccumulation. To put this into perspective, this is 1% of the total modeled river influx into the North Sea, and thus does not significantly alter the long-range transport of Hg.

**The effect of the ecosystem on the Hg budget under idealized circumstances**

The 1D setup budget is shown in Fig. 7. This shows the modeled mean tHg and tMeHg concentrations, the difference in the run without an ecosystem compared to the setup with an ecosystem, and the differences caused by biogenic reduction in the Gotland Deep.

**The Southern North Sea** setup is permanently mixed. Because of this, Hg can continuously reach the surface area where it can evaporate, and all aquatic MeHg is subject to photodegradation. At the same time, this constant mixing brings the plankton
biomass and its bioaccumulated Hg to the benthic boundary layer, where it can be consumed by macrobenthos. This results in a large flux of $Hg^{2+}$ and $MMHg^+$ from the pelagic to the benthic during the phytoplankton bloom, which is re-released during winter. This leads to high benthic Hg (11.1 pmol $m^{-2}$) and MeHg (1.6 pmol $m^{-2}$), but a relatively low burial rate (1.1 pmol tHg $m^{-2}$ $y^{-1}$) as the Hg is bioaccumulated in macrobenthos rather than associated with organic carbon of sediment. The increase in tMeHg in the Southern North Sea is the highest of the 3 setups, due to the high biomass and the ecosystem increases tMeHg
by 44%.

**The Northern North Sea** setup is only seasonally stratified. This means that during summer, macrobenthos cannot feed directly from the plankton bloom, but Hg-containing organic material can sink below the mixed layer, where it can settle as sediment, be consumed by macrobenthos, or remain until it is remixed to the surface during winter. This results in a low benthopelagic flux of $Hg^{2+}$ and $MMHg^+$ throughout the year. However, the same average Hg burial as in the Southern North
Sea of 1.1 pmol tHg $m^{-2}$ $y^{-1}$. Due to the lower biomass, there is less tMeHg in this setup than in the Southern North Sea, and the effect of the ecosystem causes a smaller increase in tMeHg (13%) than in the Southern North Sea.

**The Gotland Deep** setup has permanent stratification. Hg can accumulate in the plankton in the surface layer or divide to detritus. When this detritus sinks, it will transport Hg to deeper water. As the Gotland Deep setup is permanently stratified, Hg eventually reaches the sediment and becomes buried. The flux of pelagic $Hg^{2+}$ and $MMHg^+$ into the sediment is low, but since
there is very low resuspension, the burial rate is high (11.7 pmol tHg $m^{-2}$ $y^{-1}$) compared to the lower burial rate (1.1. pmol tHg $m^{-2}$ $y^{-1}$) for both North Sea setups. In the Gotland Deep setup, there is a big distinction between the effect of the ecosystem above and below the oxycline. Above the oxycline, the ecosystem increases tMeHg by 13%, while below the oxycline this increase is replaced by a small decrease of 3%, as the binding of $Hg^{2+}$ and $MMHg^+$ to the sinking detritus facilitates the flux of Hg to the sediment.

These differences led to the highest burial of tHg in the Gotland Deep, followed by the Northern North Sea, and the lowest in the Southern North Sea. This shows how ecosystem-induced Hg burial is influenced by local hydrodynamics.

## 4.2 The cyanobacterial reduction of $Hg^{2+}$

The model estimates that biogenic reduction can decrease average water column Hg above the mixed layer depth by 7% in the Gotland Deep, due to the transfer of soluble $Hg^{2+}$ to volatile $Hg^0$, which increases the evaporation of Hg out of the
705 water to the atmosphere. In Fig. 7 we can see that there is still an increase in tMeHg (13%) above the oxycline without bioaccumulation and biogenic reduction, while there is no increase (<0.5%) below the oxycline. This means that biogenic reduction decreased tMeHg, but the increase caused by bioaccumulation is higher above the mixed layer depth. Figure 8 shows the seasonal difference in the tHg and tMeHg in the 3D MERCY v2.0 model between runs with and without biogenic reduction and the cyanobacterial biomass in the surface layer. Note that we isolated the effect of biogenic reduction in these setups so
bioaccumulation does still occur. The MERCY v2.0 model shows a similar decrease in the Gotland Deep as the 1D GOTM setups, but it shows a higher decrease in tHg in coastal areas of up to 20%. Additionally, it shows that while cyanobacteria

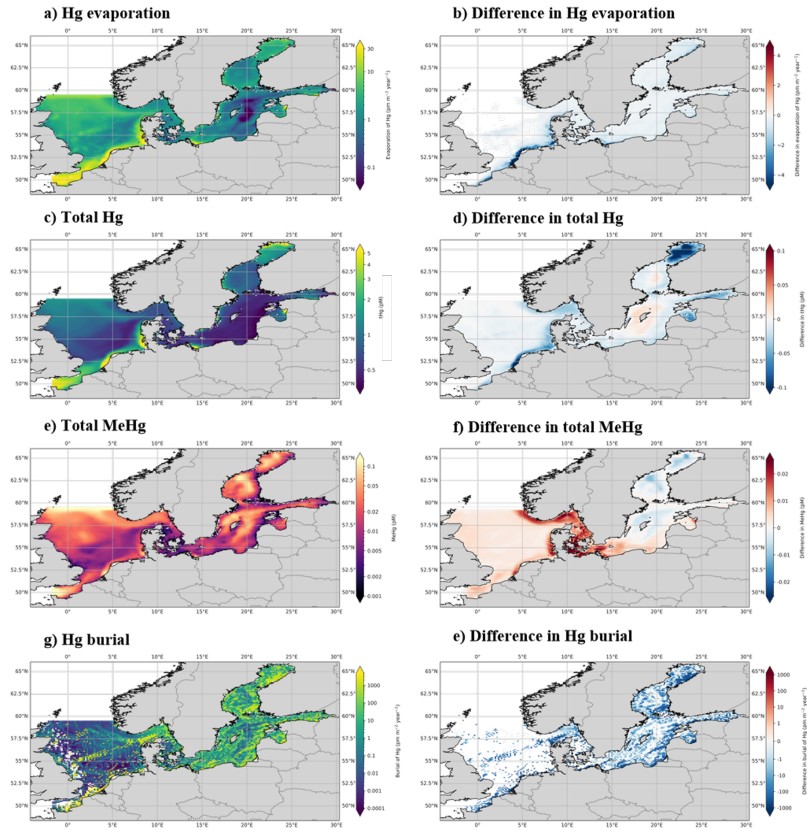

**Figure 6.** The yearly mean Hg budget of the North and Baltic Seas and the difference caused by the ecosystem, the concentrations and differences of tHg and tMeHg are the mean of the water column. The difference is calculated as the Base case - Scenario A "No bioaccumulation". The average tHg is 0.5 pM, which is decreased by 0.02 pM without bioaccumulation. The average tMeHg is 0.03 pM, which is decreased by 0.003 pM without bioaccumulation. The decrease in dissolved tHg and tMeHg without bioaccumulation is coupled with a decrease in evaporation and burial. Evaporation decreases from 4.3 pm m$^{-2}$ y$^{-1}$ to 3.9 pm m$^{-2}$ y$^{-1}$ and burial from 323.5 pm m$^{-2}$ y$^{-1}$ to 263.2 pm m$^{-2}$ y$^{-1}$.

are most abundant during autumn (Fig. 8 g), they decrease the tHg content throughout the year (Fig. 8 b, e, h, k), there is an average decrease of -16%, which is highest during the autumn bloom (-17%) and lowest in summer before the bloom (-15%). Finally, cyanobacteria and biogenic reduction only occur in the Baltic Sea in the model, but we can see a decrease in tHg in the Southern North Sea of up to -7.5 in summer and autumn and -5% in winter and spring. The difference in tMeHg is, however, less pronounced with a decrease of up to -3% in tMeHg during autumn and winter with a smaller decrease of -1% during spring and summer in the Gotland Deep. There appears to be no noticeable decrease in tMeHg in the Southern North Sea ($<$-1%). This demonstrates that cyanobacteria can have a very large impact on the tHg budget of the Baltic Sea, even in areas where they are less abundant, and this effect is relevant throughout the year. Kuss et al. (2017) finds that cyanobacterial-induced biogenic reduction causes approximately 30% of all Hg evaporation during summer, since we have cyanobacteria in our model for 3 months, a year-round average decrease of -9% of tHg above the mixed layer depth and a total decrease of up to -20% during summer and autumn is in line with these observations. Since Hg evaporation equilibrates the ocean with the atmosphere, the annual average flux is not changed dramatically (-0.3%, or -0.42 nmol m$^{-2}$ y$^{-1}$) as seen in the Fig. 7. Rather, the aquatic Hg concentration that leads to this evaporation is lower, due to a higher Hg$^{0}$/Hg$^{2+}$ ratio. In addition to the decrease of aquatic Hg, it also changes the seasonality of the evaporation of Hg$^{0}$. The cyanobacteria cause an increase in evaporation of Hg$^{0}$ during late summer and autumn facilitated by the cyanobacterial bloom, which is compensated by a small decrease in evaporation when the bloom is over, resulting in a similar yearly average evaporation of Hg$^{0}$. The cyanobacteria-facilitated reduction in tHg in our model does not mean that an abundance of cyanobacteria would automatically lead to a decreased Hg concentration in other regions. Cyanobacteria are a very diverse group, and not all cyanobacteria will decrease Hg$^{2+}$ to Hg$^{0}$ (Kuss et al., 2015). This is visualised in more detail in Fig. S4, where we show the 10-year average daily evaporation and 30-day running average of the relative difference between the 10-year average of the base case and scenario C (no bioaccumulation nor biogenic reduction).

## 4.3 Partitioning to detritus and DOM

In Fig. 9, we show that the partition of Hg$^{2+}$ and MMHg$^{+}$ into detritus and DOM causes an increase in tHg and tMeHg above the mixed layer depth in every setup. Since both North Sea setups are mixed during winter, this leads to a small increase in tHg (1-2%) in both North Sea setups. In the Gotland Deep, we observe the same increase in tHg caused by the partitioning of Hg to detritus and DOM above the mixed layer depth. However, simultaneously, we notice that partitioning to detritus and DOM decreases tHg below the mixed layer depth. This occurs because the detritus and the DOM can remove Hg from the water column since the Hg bound to the detritus can settle as sediment. Since atmospheric Hg wet deposition and atmospheric concentrations are consistent across all setups, evaporation and burial will equilibrate to the atmospheric influx. If the fraction of Hg$^{0}$ of tHg is lower, tHg will increase until Hg$^{0}$ is high enough that the evaporation of Hg$^{0}$ reaches equilibrium with atmospheric inputs. Below the mixed layer in Gotland Deep, partitioning to detritus and DOM decreases tHg. This indicates that below the mixed layer depth, the effect of the detritus and DOM partitioning on the increase in burial is stronger than its effect on the decrease in evaporation.

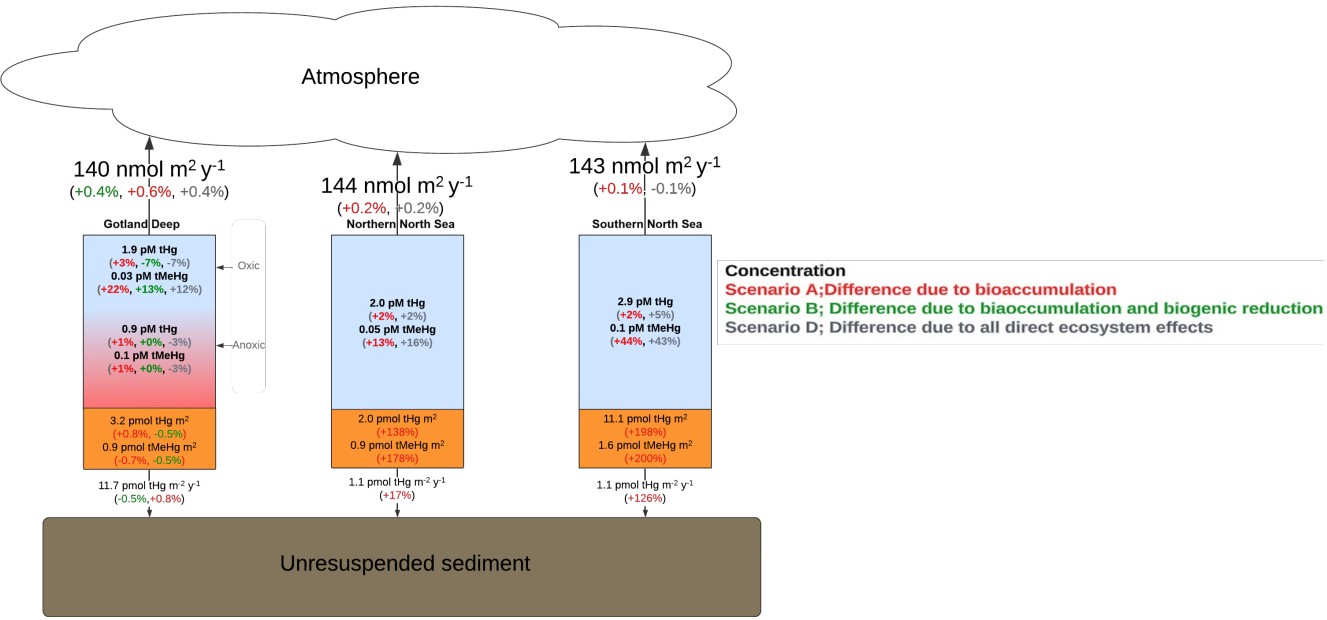

**Figure 7.** Hg Budget of the three Setups. All values are the average values for the final 10 years of the simulation. The blue area indicates oxic water, the red indicates anoxic water, and the brown area indicates sediment. The arrows indicate the direction of the flux. The black values are for the full model that includes all ecosystem effects, and the red indicates the percentage difference in the run without any ecosystem effects. The percentage differences are calculated by (Base case-scenario)/((Base case+scenario)/2). Thus negative numbers correspond to a decrease caused by the ecosystem while positive numbers indicate an increase caused by the ecosystem. In the Gotland Deep, dark green percentages show differences between the full model and setups without biogenic reduction and bioaccumulation. There is net evaporation in all setups of 140-143 nmol m$^{-2}$ y$^{-1}$. In the North Sea, there is a decrease of both tHg and tMeHg when the ecosystem is not included. This decrease is larger for tMeHg (10-39%) than for tHg (2-7%). There is a decrease of 7% of tHg in the oxic layer of the Gotland Deep caused by cyanobacterial biogenic reduction. There is a lower increase in tMeHg (12%) in the oxic water column of the Gotland Deep compared to the northern (+16%) and southern (+43%) North Sea due to all direct ecosystem effects. The effect of biogenic reduction only causes a <0.5% decrease for tHg and tMeHg in anoxic water respectively in the Gotland Deep and a 0.5% decrease in sediment and burial. Sediment concentrations of both tHg and tMeHg are lowest in the Northern North Sea, followed by the Southern North Sea, and highest in the Gotland Deep.

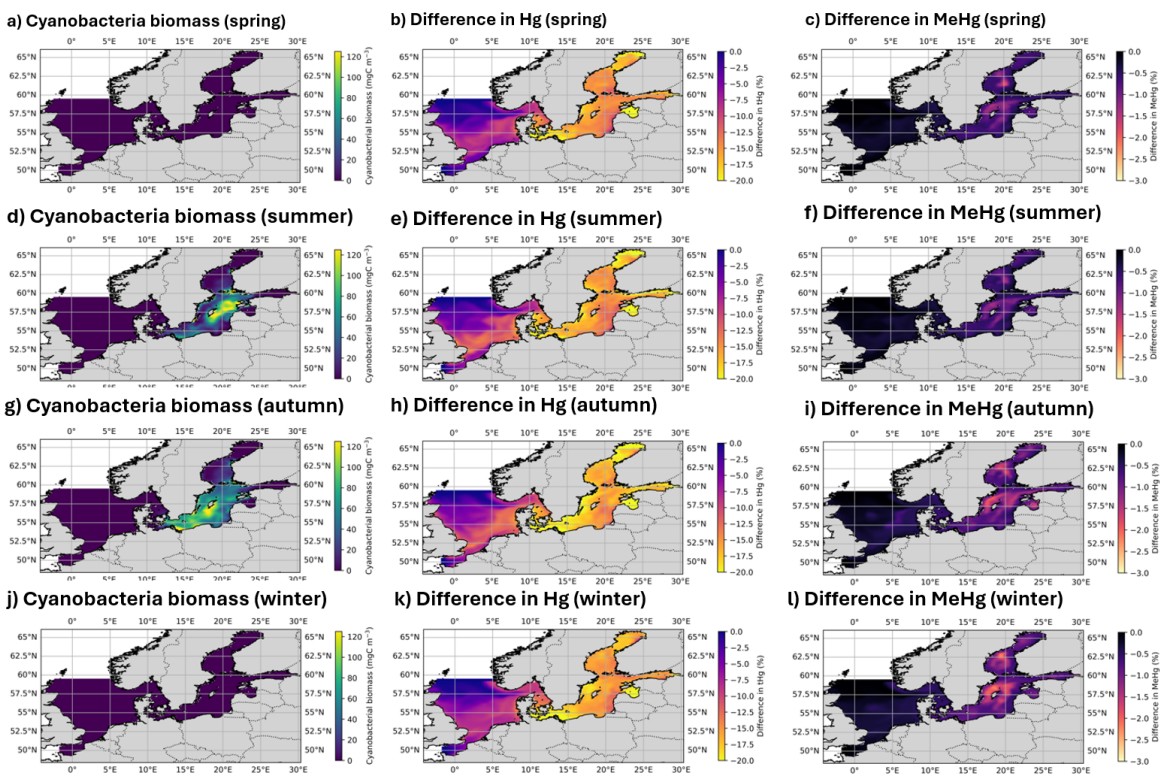

**Figure 8.** Cyanobacterial biomass and seasonal differences in tHg and tMeHg between the base case and the scenario without biogenic reduction.

## 4.4 The unique properties of Hg as drivers of these results

These results are partially driven by the unique properties of Hg as a pollutant. Because Hg as an element is stable but can be present in both methylated and nonmethylated forms, it behaves radically differently than a pollutant that would have a constant total concentration. We find that this is the case for two reasons. First, the concentration of $Hg^0$ in the surface layer is in constant exchange with the atmosphere. If the fraction of Hg present as $Hg^0$ is decreased, because $Hg^{2+}$ is bound to the biota, the evaporation of $Hg^0$ will be decreased until a new equilibrium with a higher tHg content is reached. Secondly, when

$MMHg^+$ is removed from the water column due to bioaccumulation, only a small amount of tHg is removed ($< 3\%$), which means that methylation of $Hg^{2+}$ into $MMHg^+$ is decreased by only a small amount. This would favor the net production of $MMHg^+$ which leads to an increase in tMeHg.

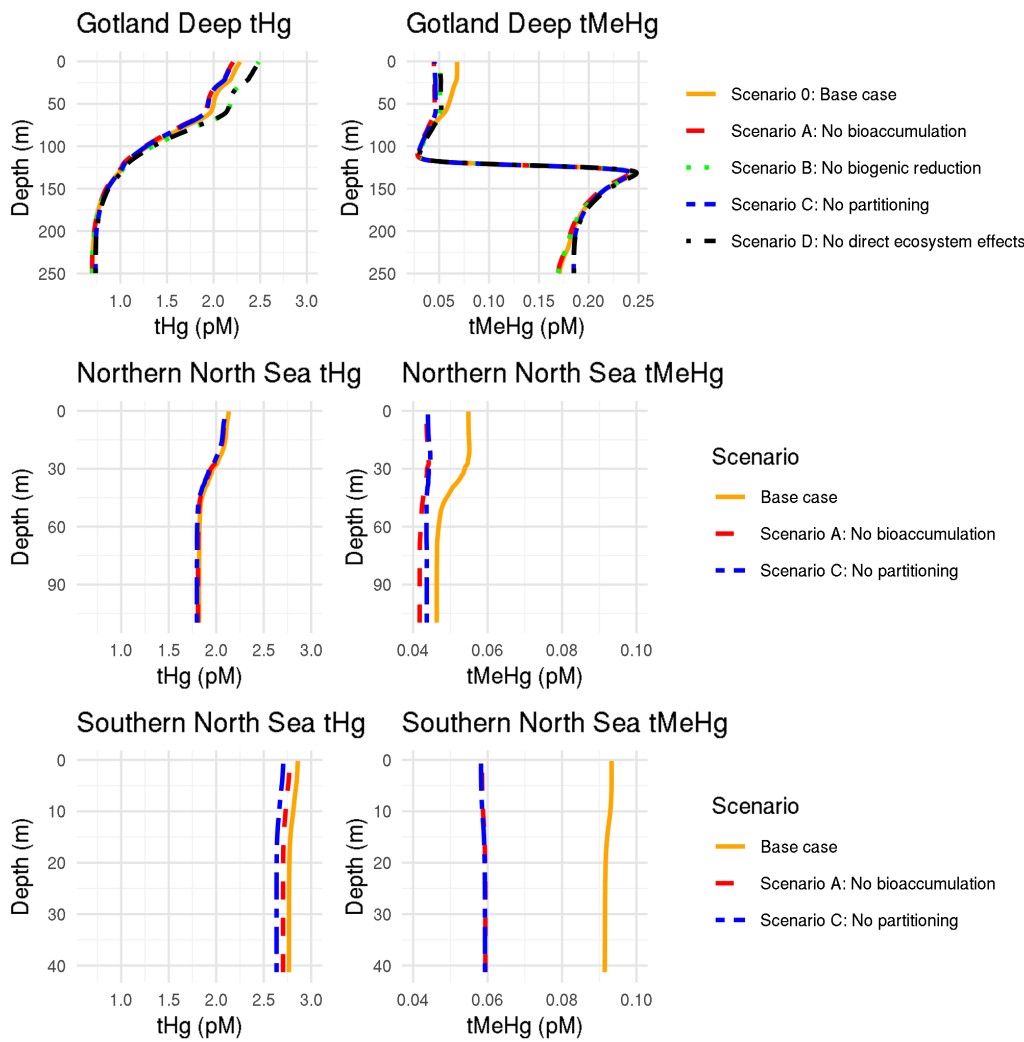

**Figure 9.** Depth profiles of tHg and tMeHg in the three setups with different model scenarios. In the Gotland Deep, the scenario without biogenic reduction (green) has an increased tHg content. This shows that biogenic reduction can decrease the yearly average water column Hg when cyanobacteria are present for part of the year. Setups without binding to detritus and DOM (blue) have lower tHg in the surface layer. This effect is strongest in the North Sea. Setups without binding to detritus and DOM (blue) also have decrease tMeHg in oxic water, but the effect is smaller than for tHg. In the Gotland Deep anoxic water, the only noticeable effect is caused by binding to detritus and DOM. In this setup, without this, there is both more tHg and tMeHg in the anoxic deep water. In all setups, runs with bioaccumulation (orange) have more tMeHg than without bioaccumulation (red). The Southern North Sea has a slightly lower amount of tHg than the Northern North Sea and Gotland Deep setups.

# 5 Model limitations

## 5.1 Movement of higher trophic level

In our model, phytoplankton and zooplankton are treated as tracers without implicit movement, while fish have no movement at all. This means that plankton and its bioaccumulated $Hg^{2+}$ and $MMHg^+$ are transported by currents but do not move themselves, while fish remain stationary. For most biological variables, this implementation is a reasonable simplification. However, in the case of fish, it might influence the model results. Moving fish could transport Hg; if fish move around, they could accumulate Hg in areas with high Hg while releasing it in areas with lower Hg, thus spreading the Hg around. Implementing migration

for fish provides an interesting direction for further model development, but in the current implementation, it would likely not cause major differences. This is because migration would only be relevant to implement in the 3D setup, but this setup is focused on plankton and only includes mid trophic levels of fish, which are less migratory than larger higher trophic level animals.

## 5.2 Decreased complexity in detritus-Hg interactions

Another limitation in the model is the limited complexity of the detritus-Hg interactions. In our model, there is only one functional group for detritus. When biota die, they instantly release all bioaccumulated $Hg^{2+}$ and $MMHg^+$, which is then in constant equilibrium with the detritus. The sinking rate of detritus bound Hg is estimated based on how much Hg is bound to detritus on a given time step. This approach is a reasonable simplification for small biota, such as phytoplankton and microzooplankton. But it introduces two limitations in higher trophic levels. First of all, if a larger animal such as fish dies and

sinks, it might transport some of its bioaccumulated Hg with the carcass. That is not accounted for in the model. Additionally, the equilibrium between detritus bound and dissolved Hg is based on small particulate organic carbon, as this is the most common form of detritus in marine water. But in the model, predators consume detritus, and the detritus consumed by higher trophic level predators would mostly be larger particles that could have different Hg binding characteristics.

## 5.3 Limitations in the modeled ecosystem complexity

As discussed when analyzing the modeled and observed trophic levels, the ecosystem model uses a functional group approach to constrain the complexity of the ecosystem. This does, however, introduce limitations in the model's ability to model bioaccumulation. Several animals might be classified under the same functional group while affecting Hg dynamics in different ways. An example is the observations that some Baltic Sea zooplankton can have *in vivo* Hg speciation (Gorokhova et al., 2020). Increasing the modeled ecosystem complexity could improve the model's performance by accounting for these differences.

## 5.4 Physical drivers of bioaccumulation

In the current parametrization, physical drivers such as temperature and light influence the biomass and the Hg speciation, but the bioaccumulation is purely based on the concentration of bioaccumulative Hg species, a biota functional group specific

bioaccumulation rate, and the biomass of the biotic functional group. The only temperature dependent driver directly influencing bioaccumulation is the temperature dependent respiration rate of fish. As the temperature increases, so does the respiration rate of fish and consequently they release Hg faster, as this is coupled to their respiration rate. As shown by Garcia-Arevalo et al. (2024) the bioaccumulation of $MMHg^+$ in phytoplankton is also dependent on cell dependent drivers, such as the availability of membrane transport channels. These cell dependent drivers might be different during different stages of the phytoplankton bloom due to altering bloom composition or the physiological state of phytoplankton. While these changes could not be incorporated in our current model due to lack of our understanding of nuanced drivers of bioaccumulation, seasonal changes could influence bioaccumulation at every trophic level.

## 6   The required ecosystem complexity to capture Hg dynamics

As discussed, a main conclusion of this paper is showing that the ecosystem is an integral part of Hg cycling, and that this should not be overlooked. However, there is nuance in how the ecosystem should be implemented in Hg cycling models, and there is a trade-off between keeping the model simple and ensuring key drivers are implemented.

### 6.1   High trophic levels as a reservoir of MeHg

In most marine ecosystems, the annual average biomass of primary producers is relatively low; rather, there is a very high turnover rate of primary producers during the bloom period. While the exact numbers vary depending on the location and seasonality, high trophic levels can make up a major component of the total ecosystem biomass, especially in winter. As high trophic levels have the most $MMHg^+$ per biomass, they can form a major reservoir of $MMHg^+$. Our model, however, shows that this does not have a major effect on the tHg concentration. This indicates that the inclusion of high trophic levels such as fish might be necessary to correctly estimate the $MMHg^+$ budget, but the inclusion of fish is not necessary to correctly model tHg fluxes. One point of uncertainty here is that this conclusion is based on our implementation of the ecosystem. As discussed in the model limitation segment, several drivers, such as fish migration and the transport of Hg to deep water by sinking carcasses, are not accounted for, and these drivers could still prove to be an essential component of Hg cycling.

### 6.2   Bentho pelagic coupling

A key component where the ecosystem is essential for a correct understanding of Hg cycling is the bentho-pelagic coupling. In coastal areas, the consumption of pelagic detritus and Hg bound to it by macrobenthos can be a major flux of organic carbon from the pelagic to the benthic system. The sediment is identified as a key area for Hg methylation; this increased transport of $Hg^{2+}$ from the pelagic to the benthic due to biotic consumption of detritus would constitute a source of $MMHg^+$. Of course, in some areas, sediment is not resuspended, and then increased transport of Hg to the sediment can result in additional burial of Hg. As the macrobenthic influence on the bentho-pelagic coupling, the burial and resuspension are spatially and temporally variable; this cannot be accounted for by a "back of the envelope" estimation. Rather, the inclusion of a realistic bentho-pelagic coupling is essential for Hg speciation models in coastal areas.

## 6.3 key biota for Hg cycling

In an important aspect of estimating the role of the ecosystem in Hg cycling is understanding that not all biota affect Hg cycling in a similar way. The clearest of the ecosystem interactions with Hg cycling is the removal of Hg by biota when biomass is high and the release when biomass is low. This is extensively analyzed in this study. However, several biota might have an unexpectedly high impact on Hg cycling. The first example of this, which is also evaluated in this paper, is Baltic Sea cyanobacteria, which can facilitate biogenic reduction. But beyond this, it is demonstrated that some animals of zooplankton and cephalopods can have *in vivo* Hg speciation (Gente et al., 2023; Gorokhova et al., 2020). Another example is that sponges are demonstrated to have very high inorganic Hg levels, suggesting an important role in the bentho-pelagic coupling (Orani et al., 2020).

## 7 Summary

In this study, we hypothesized that the ecosystem is an essential part of marine Hg cycling. We quantified the impact of the ecosystem on marine Hg cycling by simulating a 1D water column with and without bioaccumulation, biogenic reduction, and partitioning into detritus and dissolved organic matter (DOM). Furthermore, we ran a 3D model for the North and Baltic Seas with and without bioaccumulation to analyze the spatial effects. Our analysis focused on the effects of these ecosystem interactions on total Hg (tHg) and total methylated Hg (tMeHg) concentrations in the water column and the intercompartmental fluxes.

Our model demonstrates the complex differences in bioaccumulation between different species of Hg. The model accurately reproduces bioaccumulation at the base of the food web as shown in Table 4 and models biomagnification to higher trophic levels according to observations for both $Hg^{2+}$ and $MMHg^+$ as shown in Fig. 4. Although our model underrepresents $MMHg^+$ in cod compared to the observed mean, this underrepresentation is still within the observations and is explained by an underrepresentation of the trophic level as shown in Table A2. It is important to note that we wanted to implement realistic bioconcentration and trophic transfer rates to not overtune the model. Several interactions, such as, for example, cannibalism within the functional groups, can increase bioaccumulation in ways that are not captured by the model, which would result in both increased bioaccumulation and trophic levels.

The impact of the ecosystem on the MeHg cycling is very strong. In idealized 1D setups, bioaccumulation increases the average tMeHg content by 44% in the Southern North Sea, by 13% in the Northern North Sea, and in the Gotland Deep above the mixed layer depth by 22% as shown in Fig. 7, note that the percentage difference due to ecosystem drivers can deviate from the earlier presented difference in the surface layer. The surge in tMeHg attributed to bioaccumulation is most notable during plankton blooms, where phytoplankton absorbs a large portion of tMeHg, shielding bioaccumulated $MMHg^+$ from demethylation processes. This $MMHg^+$ is released to the water column towards the end of the year, where less solar radiation, more particles, more mixing, and more cloud coverage decrease photodemethylation. Our models show that the increase in tMeHg is strongly related to average biomass. An increase of 4.5 mgC m$^{-3}$ in the average biomass content will result in a 1% increase in tMeHg due to bioaccumulation.

The model reveals regional and seasonal differences in how the ecosystem influences Hg cycling and bioaccumulation across the North and Baltic Seas. The ecosystem increases tHg and tMeHg mostly in highly productive shallow coastal regions, with a decreased effect in deeper, less productive zones. In the Baltic Sea, our findings quantify the average decrease in tHg due to bioaccumulation and cyanobacterial-induced biogenic reduction as a 7% decrease in tHg above the mixed layer depth.

The 3D simulation expands on this by visualizing the spatial pattern and allowing us to quantify the effect of bioaccumulation on the Hg budget of the North and Baltic Seas while taking into account spatial variability and horizontal transport. The average tMeHg content increases by 10%, but this increase is caused by an increase in the North Sea and the Danish Straits, while there is a decrease in tMeHg in large parts of the Baltic Sea.

## 7.1 Conclusion

This study demonstrates and quantifies the complex role of the ecosystem in shaping Hg speciation and the influence of physical, biochemical, and ecological factors. In addition, it shows the sensitivity of parameterization in modeling and shows that evaluating ecosystem parameters such as the trophic level is essential to comprehend the results of bioaccumulation modeling. Here, we show that bioaccumulation does have notable feedback effects on Hg cycling and therefore should be included in any marine Hg model, even in cases where bioaccumulation is not of direct interest. We conclude that the ecosystem has a direct effect on Hg cycling by;

- The ecosystem increases marine tMeHg

    - Increase in tMeHg up to 77%

    - Increase of the 10-year average tMeHg up to 44%

    - An increase of 4.5 mgC $^{-3}$ average biomass leads to a 1% increase in tMeHg.

- Facilate burial of Hg by transporting Hg to below the thermocline in deep unmixed water via the sinking of detritus

- Cause a wintertime increase in both aquatic and tMeHg permanently mixed productive coastal water

Because of this, we conclude that the ecosystem has essential feedback on marine Hg cycling.

## 8 Future outlook

It is important to continue improving our models and to increase our understanding of the mechanisms that drive Hg bioaccumulation and cycling. More efforts should be directed toward understanding the nuanced interactions between Hg cycling and biogeochemical processes, trophic interactions, and ecosystem structure, all of which influence Hg speciation and fate in marine ecosystems. Typically, bioaccumulation is thought of as an add-on to model additional end members, and thus as non-essential when modeling global transport and air-sea exchange. We demonstrate that this is not the case and that bioaccumulation plays an important role in Hg cycling in coastal oceans.

To support the Minamata Convention, a solid understanding of Hg cycling in marine environments is important for designing effective management strategies that aim to mitigate Hg contamination and minimize its impact on aquatic ecosystems and human health. By improving the models and by incorporating new advancements in field observations and experimental studies, we can increase our ability to predict and manage Hg pollution, safeguarding both the environment and human well-being.

## Conflict of interest

None of the authors declare any competing interests.

### Funding

This research has been funded by the European Union's Horizon 2020 research and innovation programme under the Marie Sklodowska-Curie grant agreement no. 860497.

## Author contributions

The authors contributed to the article as stated in Table 7.

### Acknowledgement

AI assisted spell check was used in Grammarly and Writefull, while readability suggestions were occasionally provided by an rAI model such as chatGPT (OpenAI). In addition, AI tools helped optimize R and Python visualizations. All suggestions were critically evaluated and implemented only after manual verification. All final text is written and verified by the authors. The sources were searched with Perplexity AI or Google Scholar, but all sources were manually read, verified, and cited.

**Table 7.** Contributions per Author. Authors are: David Johannes Amptmeijer (DA), Dr. Johannes Bieser (JB), Dr. Ute Daewel (UD), Elena Mikheeva (EM), and Prof. Dr. Corinna Schrum (CS).

| Contributor role | Role definition | Authors |
|---|---|---|
| Conceptualisation | Identified the need for bioaccumulation in the MERCY v2.0 model | JB, CS |
| | Conceptualised the study | DA, JB, CS |
| Methodology | Coupling of MERCY v2.0 to FABM | DA, JB |
| | Developing the physical setups | EM |
| | Developed ECOSMO E2E to better suit bioaccumulation | DA, UD |
| | Design and implement the different scenarios | DA |
| Validation | Validate if Hg cycling matches in the 1D MERCY v2.0 model | JB, DA |
| | Validate hydrodynamic conditions | EM, CS, DA |
| | Validate carbon cycling | DA, UD |
| Writing | Writing of the original draft | DA, JB |
| | Reviewing the original draft and quality control | JB, CS, UD, EM, DA |
| Supervision | Supervising the development of the work | CS, JB |
| Funding acquisition | Acquired funding via the GMOS-Train ITN | JB |

## Appendix A:  Appendix 1: Model parameterization

**Table A1.** Dimensions, shape, and maximum growth and mortality rates of the phytoplankton model organism in the North and Baltic Seas, to resemble ECOSMO E2E functional groups and the conversion ratio of mg C to $cm^2$ cell membrane and $dm^3$ cell volume. The dimensions and shapes are based on Olenina et al. (2003). For the Cylinder they are radius and height, for the sphere and hemisphere - the radius. The maximum growth rate and the mortality are based on Daewel et al. (2019).

| Group | Organism | Shape | Dimensions ($\mu$m ) | $\mu_{max}$ ($d^{-1}$) | Mortality ($d^{-1}$) |
|---|---|---|---|---|---|
| Diatoms | *T. baltica* | Cylinder | 12.5 x 25 | 1.4 | 0.04 |
| Flagellate | *P. Catanata* | Hemisphere | 6 | 1.2 | 0.04 |
| Cyanobacteria | *A. flos-aquae* | Sphere | 4 | 1.0 | 0.08 |

**Table A2.** The parameterization of consumers. The preference and consumption rate ($r_{cons}$) determine how much of each prey is consumed, the assimilation efficiency (AE) how much of the consumed carbon is assimilated into the predator; the mortality ($r_{mort}$) and respiration ($r_{resp}$) rate determine the total loss. Adjustments were made for higher trophic levels compared to Daewel et al. (2019) to enhance the model's suitability for bioaccumulation. Specifically, the AE and grazing rate were lowered. Additionally, fish 2 was introduced as a top predator with modified parameters. It has a higher preference for macrobenthos and consumes fish 1 rather than microzooplankton. As a top predator, fish 2 has a lower AE and consumption rate compared to fish 1. All other rates and equations remain consistent with the ECOSMO E2E model (Daewel et al., 2019).

| Group | Prey | Preference (1) | $r_{cons}$ (d$^{-1}$) | AE (1) | $r_{mort}$ (d$^{-1}$) | $r_{resp}$ (d$^{-1}$) |
|---|---|---|---|---|---|---|
| Microzooplankton | Diatom | 0.25 | 0.8 | 0.75 | 0.05 | 0.02 |
| | Flagellates | 0.70 | 0.8 | 0.75 | | |
| | Cyanobacteria | 0.30 | 0.3 | 0.75 | | |
| | Detritus | 0.10 | 0.8 | 0.8 | | |
| Mesozooplankton | Diatom | 0.85 | 0.7 | 0.6 | 0.025 | 0.015 |
| | Flagellates | 0.10 | 0.7 | 0.6 | | |
| | Cyanobacteria | 0.30 | 0.3 | 0.6 | | |
| | Microzooplankton | 0.15 | 0.8 | 0.6 | | |
| | Detritus | 0.10 | 0.7 | 0.6 | | |
| Macrobenthos | Phytoplankton | 0.2 | 0.1 | 0.6 | 0.01 | 0.001 |
| | Zooplankton | 0.2 | 0.2 | 0.6 | | |
| | Sediment | 0.15 | 0.15 | 0.6 | | |
| | Detritus | 0.15 | 0.15 | 0.6 | | |
| | DOM | 0.15 | 0.1 | 0.6 | | |
| Fish 1 | Microzooplankton | 0.45 | 0.015 | 0.5 | 0.001 | 0.002 |
| | Mesozooplankton | 0.45 | 0.015 | 0.5 | | |
| | Macrobenthos | 0.05 | 0.015 | 0.5 | | |
| | Detritus | 0.05 | 0.0125 | 0.5 | | |
| Fish 2 | Mesozooplankton | 0.25 | 0.012 | 0.45 | 0.001 | 0.002 |
| | Fish 1 | 0.25 | 0.012 | 0.45 | | |
| | Macrobenthos | 0.45 | 0.013 | 0.45 | | |
| | Detritus | 0.05 | 0.001 | 0.45 | | |

**Table A3.** The estimated bioconcentration ($r_{bc}$), release rates ($r_{rel}$), and turnover rates ($r_{to}$) for phytoplankton (Mason et al., 1996), zooplankton (Tsui and Wang, 2004) and fish (Wang and Wong, 2003; Trudel and Rasmussen, 1997). Bioconcentration rates are in $m^3$ mg C $^{-1}$ $d^{-1}$ and release and turnover rates are $d^{-1}$ (Trudel and Rasmussen, 1997; Pickhardt et al., 2006; Mason et al., 1996; Tsui and Wang, 2004). The bioconcentration rates for phytoplankton are based on the cell size, for zooplankton and macrobenthos on *Dapnia pulex*, and for fish on *Plectorhinchus gibbosus*.

| Functional group | $Hg^{2+}$ | | | $MMHg^+$ | | |
|---|---|---|---|---|---|---|
| | $r_{bc}(m^3 mgC d^{-1})$ | $r_{rel}(d^{-1})$ | $r_{to}(d^{-1})$ | $r_{bc}(m^3 mgC d^{-1})$ | $r_{rel}(d^{-1})$ | $r_{to}(d^{-1})$ |
| Diatoms | 3.2E-3 | 63.1 | (-) | 3.1E-3 | 0.75 | (-) |
| Flagellates | 3.2E-3 | 65.5 | (-) | 3.1E-3 | 0.75 | (-) |
| Cyanobacteria | 5.5E-3 | 109.7 | (-) | 5.4E-3 | 0.75 | (-) |
| Microzooplankton | 1.68E-5 | 0.03 | 0.03 | 2.22E-05 | 7.50E-01 | 7.50E-01 |
| Mesozooplankton | 1.68E-5 | 3.1E-3 | 3.1E-3 | 2.22E-05 | 1.50E-02 | 1.50E-02 |
| Macrobenthos | 1.68E-5 | 0.04 | 0.04 | 2.22E-05 | 2.50E-02 | 2.50E-01 |
| Fish 1 | 3.90E-7 | 2.16E-2 | 4.47E-2 | 9.07E-6 | 2.90E-03 | 3.00E-04 |
| Fish 2 | 3.90E-7 | 2.16E-2 | 4.47E-2 | 9.07E-6 | 2.90E-03 | 3.00E-04 |

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
