# Peer review of "Bioaccumulation as a driver of high MeHg in the North and Baltic Seas"

_EGUsphere, 2025_

## Author Comment (AC1)

**1  Mayor comments**

**Reviewer Comment**

1) While I recognize that this is a complex model and study design, the manuscript is on the long end, and there may be opportunities to streamline the text (particularly in 2.2-2.7) to avoid repetition and allow the core messages to come through more clearly. The authors could consider using a Supplemental Information Section to present some details of the model assumptions and parameterization, as well as for some supporting figures (e.g., Figure 12). In addition, the manuscript would benefit from additional review for typos and readability. Some have been flagged below in minor comments.

**Author Response**

Thank you for your insights. I suggest that I move the model specifics such as the exact parameterization to the supplementary section. I suggest that I move Table 2 and Table 3 to the supplement. It is essential to show the exact parameterization used, but the large tables distract from the core message of the paper. Lastly, I suggest moving Figures 4, 5, and 6 to the supplement. I think it is good to show the seasonal progression of the bioaccumulation in all setups, but moving it to the supplement would allow an interested reader to verify the seasonal progression of the bioaccumulation while not distracting from the core message of the paper, especially since we have no seasonal data to verify against. Additionally, as you suggested, I would move Fig. 12 to the supplement. The results shown in Fig. 12 are interesting and help contextualize the air-sea exchange, but they are indeed not a core part of the message. Putting that in the supplement will mean it does not distract and bloat the paper. The tables and figures below I would suggest moving to the supplement:

**Table 2.** Dimensions, shape, and maximum growth and mortality rates of most common phytoplankton species in the North and Baltic Seas, to resemble ECOSMO E2E functional groups and the conversion ratio of mg C to $cm^2$ cell membrane and $dm^3$ cell volume. The dimensions and shapes are based on Olenina et al. (2003). For the Cylinder they are radius and height, for the sphere and hemisphere - the radius. The maximum growth rate and the mortality are based on Daewel et al. (2019).

| Group | Species | Shape | Dimensions ($\mu m$) | $\mu_{max}$ ($d^{-1}$) | Mortality ($d^{-1}$) |
|---|---|---|---|---|---|
| Diatoms | *T. baltica* | Cylinder | 12.5 x 25 | 1.4 | 0.04 |
| Flagellate | *P. Catanata* | Hemisphere | 6 | 1.2 | 0.04 |
| Cyanobacteria | *A. flos-aquae* | Sphere | 4 | 1.0 | 0.08 |

**Table 3.** The parameterization of consumers. The preference and consumption rate ($r_{cons}$) determine how much of each prey is consumed, the assimilation efficiency (AE) how much of the consumed carbon is assimilated into the predator; the mortality ($r_{mort}$) and respiration ($r_{resp}$) rate determine the total loss. Adjustments were made for higher trophic levels compared to Daewel et al. (2019) to enhance the model's suitability for bioaccumulation. Specifically, the AE and grazing rate were lowered. Additionally, fish 2 was introduced as a top predator with modified parameters. It has a higher preference for macrobenthos and consumes fish 1 rather than microzooplankton. As a top predator, fish 2 has a lower AE and consumption rate compared to fish 1. All other rates and equations remain consistent with the ECOSMO E2E model (Daewel et al., 2019).

| Group | Prey | Preference (1) | $r_{cons}$ ($d^{-1}$) | AE (1) | $r_{mort}$ ($d^{-1}$) | $r_{resp}$ ($d^{-1}$) |
|---|---|---|---|---|---|---|
| Microzooplankton | Diatom | 0.25 | 0.8 | 0.75 | 0.05 | 0.02 |
| | Flagellates | 0.70 | 0.8 | 0.75 | | |
| | Cyanobacteria | 0.30 | 0.3 | 0.75 | | |
| | Detritus | 0.10 | 0.8 | 0.8 | | |
| Mesozooplankton | Diatom | 0.85 | 0.7 | 0.6 | 0.025 | 0.015 |
| | Flagellates | 0.10 | 0.7 | 0.6 | | |
| | Cyanobacteria | 0.30 | 0.3 | 0.6 | | |
| | Microzooplankton | 0.15 | 0.8 | 0.6 | | |
| | Detritus | 0.10 | 0.7 | 0.6 | | |
| Macrobenthos | Phytoplankton | 0.2 | 0.1 | 0.6 | 0.01 | 0.001 |
| | Zooplankton | 0.2 | 0.2 | 0.6 | | |
| | Sediment | 0.15 | 0.15 | 0.6 | | |
| | Detritus | 0.15 | 0.15 | 0.6 | | |
| | DOM | 0.15 | 0.1 | 0.6 | | |
| Fish 1 | Microzooplankton | 0.45 | 0.015 | 0.5 | 0.001 | 0.002 |
| | Mesozooplankton | 0.45 | 0.015 | 0.5 | | |
| | Macrobenthos | 0.05 | 0.015 | 0.5 | | |
| | Detritus | 0.05 | 0.0125 | 0.5 | | |
| Fish 2 | Mesozooplankton | 0.25 | 0.012 | 0.45 | 0.001 | 0.002 |
| | Fish 1 | 0.25 | 0.012 | 0.45 | | |
| | Macrobenthos | 0.45 | 0.013 | 0.45 | | |
| | Detritus | 0.05 | 0.001 | 0.45 | | |

[Figure]

**Figure 4.** Hg accumulation in the Gotland Deep during the last simulation year (Jan 2010-Jan 2011). Plot 4a shows tHg bioaccumulation with mean and range of observations from Nfon et al. (2009) represented by the point and vertical bar on the right side of the plot. Bioaccumulation is displayed when biomass of the respecitive funtional group exceeds 0.1 gC m$^{-2}$. tHg bioaccumulation is highest in fish 2, followed by fish 1, microzooplankton, mesozooplankton, cyanobacteria, flagellates, and diatoms with tHg concentrations in observed ranges. The consecutive Fig. show the bioamagnification (4b, 4d) and bioconcentration (4c, 4e) of Hg$^{2+}$ (4b, 4c) and MMHg$^{+}$ (4d, 4e). Biomagnified Hg$^{2+}$ is highest in microzooplankton, followed by mesozooplankton and fish, while biomagnified MMHg$^{+}$ increases notable in fish 1 and fish 2. Bioconcentrated Hg$^{2+}$ is very low in fish and highest in zooplankton. Bioconcentrated MMHg$^{+}$ is notable higher in cyanobacteria than in all biota and lowest zooplankton.

[Figure]

**Figure 5.** Hg bioaccumulation in the Northern North Sea. Plot 5a shows the tHg bioaccumulation in the Northern North Sea. Phytoplankton and zooplankton are shown if their average biomass is more than 0.1 gC m$^{-2}$. Phytoplankton has the lowest tHg, which is followed by fish 1, fish 2, mesozooplankton and macrobenthos. Plots 5 b-e show the origin (Biomagnification or Bioconcentration) and species (Hg$^{2+}$ or MMHg$^{+}$) of the bioaccumulated tHg. Figure 5b and 5c show that the high tHg in microzooplankton, mesozooplankton and macrobenthos is due to high levels of Hg$^{2+}$ bioconcentration and biomagnification. MMHg$^{+}$ Biomagnification follows a pattern in which it is lower in zooplankton and macrobenthos, higher in fish 1, and highest in fish 2. This means that while fish 2 has a lower Hg content than macrobenthos and zooplankton for part of the year, MMHg$^{+}$ is higher in fish than in zooplankton and macrobenthos. Figure 5e shows the bioconcentration of MMHg$^{+}$ and shows that this is highest in phytoplankton, followed by fish 1 and 2, macrobenthos, and zooplankton.

[Figure]

**Figure 5.** Hg bioaccumulation in the Northern North Sea. Plot 5a shows the tHg bioaccumulation in the Northern North Sea. Phytoplankton and zooplankton are shown if their average biomass is more than 0.1 gC m$^{-2}$. Phytoplankton has the lowest tHg, which is followed by fish 1, fish 2, mesozooplankton and macrobenthos. Plots 5 b-e show the origin (Biomagnification or Bioconcentration) and species (Hg$^{2+}$ or MMHg$^{+}$) of the bioaccumulated tHg. Figure 5b and 5c show that the high tHg in microzooplankton, mesozooplankton and macrobenthos is due to high levels of Hg$^{2+}$ bioconcentration and biomagnification. MMHg$^{+}$ Biomagnification follows a pattern in which it is lower in zooplankton and macrobenthos, higher in fish 1, and highest in fish 2. This means that while fish 2 has a lower Hg content than macrobenthos and zooplankton for part of the year, MMHg$^{+}$ is higher in fish than in zooplankton and macrobenthos. Figure 5e shows the bioconcentration of MMHg$^{+}$ and shows that this is highest in phytoplankton, followed by fish 1 and 2, macrobenthos, and zooplankton.

[Figure]

**Figure 12.** The 10-year average daily atmospheric exchange of Hg0 between the atmosphere and the sea surface in the base case and scenario c (no bioaccumulation nor biogenic reduction).

**Reviewer Comment**

2) It could be helpful to provide a brief summary of the drivers of spatial and temporal variation in the results, as some of these details may be contained in the cited original model papers and therefore less clear to a reader. For example, for seasonality: to what extent is temperature dependence also considered in the bioaccumulation and toxicokinetic modeling, in addition to biomass modeling? For spatial variation: What determines the spatial distribution of higher trophic levels? Is migration relevant and, if so, how is it considered? If not, what additional implications could this have for the spatial dynamics?

**Author Response**

Thank you for your comment. I will expand on this in several ways. I will first discuss in the Mercy and Ecosmo sections respectively what spatial and temperaol drivers control biomass and Hg speciation and then add a segment to the discussion. In section 2.2.3 I would suggest to add:

**Suggested edit**

[revised manuscript text omitted]

**Reviewer Comment**

3) See below for some places where clarification of some methodological details could be beneficial, potentially in supporting material (e.g., in model-obs comparison for 1D, initial conditions).

**Reviewer Comment**

4) The authors may have the opportunity to deepen the reflection on next steps and future directions, given the importance of the call to better represent ecosystem effects in models. Some questions I am particularly curious to get their thoughts on are: a) is model coupling the only way to do this, is it reasonable to do a back-of-the-envelope adjustment factor that is regionally specific; b) how much trophodynamic complexity is needed — does capturing the base of the food web get most of the effect or do fish 1 and 2 shift the patterns; if so, what might be missing in this current simplified representation of the ecosystem

**Author Response**

I would suggest that I add the following part to the discussion segment:

**Suggested edit**

**The required ecosystem complexity to capture Hg dynamics**

As discussed, a main conclusion of this paper is showing that the ecosystem is an integral part of Hg cycling, and that this should not be overlooked. However, there is nuance in how the ecosystem should be implemented in Hg cycling models, and there is a trade-off between keeping the model simple and ensuring key drivers are implemented.

**High trophic levels as a reservoir of $MMHg^+$**

In most marine ecosystems, the annual average biomass of primary producers is relatively low; rather, there is a very high turnover rate of primary producers during the bloom period. While the exact numbers vary depending on the location and seasonality, high trophic levels can make up a major component of the total ecosystem biomass, especially in winter (Bar-On et al., 2018). As high trophic levels have the most $MMHg^+$ per biomass, they can form a major reservoir of $MMHg^+$. Our model, however, shows that this does not have a major effect on the tHg concentration. This indicates that the inclusion of high trophic levels such as fish might be necessary to correctly estimate the $MMHg^+$ budget, but the inclusion of fish is not necessary to correctly model tHg fluxes. One point of uncertainty here is that this conclusion is based on our implementation of the ecosystem.

As discussed in the model limitation segment, several drivers, such as fish migration and the transport of Hg to deep water by sinking carcasses, are not accounted for, and these drivers could still prove to be an essential component of Hg cycling.

**Benthic-pelagic coupling**

A key component where the ecosystem is essential for a correct understanding of Hg cycling is the Benthic-pelagic coupling. In coastal areas, the consumption of pelagic detritus and Hg bound to it by macrobenthos can be a major flux of organic carbon from the pelagic to the benthic system (Griffiths et al., 2017). The sediment is identified as a key area for Hg methylation; this increased transport of $Hg^{2+}$ from the pelagic to the benthic due to biotic consumption of detritus would constitute a source of $MMHg^+$ (Helmrich et al., 2022). Of course, in some areas, sediment is not resuspended, and then increased transport of Hg to the sediment can result in additional burial of Hg. As the macrobenthic influence on the benthic-pelagic coupling are spatially and temporally variable this cannot be accounted for by basic estimation. Rather, the inclusion of a realistic benthic-pelagic coupling is essential for Hg speciation models in coastal areas.

**Key species for Hg cycling**

In important aspect of estimating the role of the ecosystem in Hg cycling is understanding that not every species affects Hg cycling in a similar way. The clearest of the ecosystem interactions with Hg cycling is the removal of Hg by biota when biomass is high and the release when biomass is low. This is extensively analyzed in this study. However, several species might have an unexpectedly high impact on Hg cycling. The first example of this, which is also evaluated in this paper, is Baltic Sea cyanobacteria, which can facilitate biogenic reduction. But beyond this, it is demonstrated that some species of zooplankton and cephalopods can have *in vivo* Hg speciation (Gente et al., 2023; Gorokhova et al., 2020), and that bioaccumulation is extremely sensitive to *in vivo* Hg speciation (Li et al., 2022). Another example is that sponges are demonstrated to have very high inorganic Hg levels, suggesting an important role in the benthic-pelagic coupling (Orani et al., 2020). Identifying such key species and their effect on the Hg cycle is essential to understand there rol in the Hg cycle and if there implementation in models is needed.

**2 Detailed comments**

**Reviewer Comment**

L22: Number of parties now exceeds the number of signatories (over 150), so could update the number https://minamataconvention.org/en/parties

**Author Response**

The link provided still states that indeed has 152 parties. I suggest to update the statement as below to show both:

**Suggested edit**

Due to the consumption of polluted marine wildlife, more than 1000 people died, and more were permanently disabled (Harada, 1995). Efforts to control Hg emissions culminated in the Minamata Convention on Mercury, which is a pledge to reduce Hg emissions(Outridge

et al., 2018). It has 152 parties and is currently signed by 128 countries.

**Reviewer Comment**

L52: "In summary, there are three fractions... in our model." Read as confusing as the model hasn't been introduced yet.

**Author Response**

I agree that this is incorrect to refer to this in the model before the model is introduced. I also think the rest of the alinea could be clearer. I would suggest rewriting it as:

**Suggested edit**

Under anoxic conditions, $Hg^{2+}$ binds with $S^{2-}$ to form cinnabar (HgS), which is considered a sink due to its low solubility (Oliveri et al., 2016). In seawater, the abundance of chloride ions causes $Hg^{2+}$ and $MMHg^+$ to exist mainly in the form of inorganic chlorine complexes. The neutral forms of these complexes, $HgCl_2$ and MMHgCl, are lipophilic and can diffuse through cell membranes or bind to organic material (Zhong & Wang, 2009). The speciation of Hg with organic carbon in the marine ecosystem, such as detritus and DOM, is a complex interaction that can influence the speciation, solubility, mobility, membrane permeability, and toxicity of Hg (Ravichandran, 2004). In this study, we refer to three distinct fractions of both inorganic Hg and MMHg: 1) dissolved species not bound to organic material, including species such as $HgCl_2$ and MMHgCl, collectively referred to as $Hg^{2+}$ and $MMHg^+$, 2) Hg and MMHg bound to dissolved organic matter (DOM), referred to as Hg-DOM and MMHg-DOM, and Hg and MMHg bound to detritus, referred to as Hg-detritus and MMHg-detritus.

**Reviewer Comment**

L137: As defined in the first sentence, isn't this bioconcentration only?

**Author Response**

That is indeed badly phrased. I would suggest that I correct it to:

**Suggested edit**

*Bioaccumulation* is the increase in $Hg^{2+}$ or $MMHg^+$ in the biota relative to the concentration of the surrounding water.

**Reviewer Comment**

Fig. 1: Typos in title and Scenario C. Could consider overlaying the 1-D vs 3-D component too so that it captures that aspect of the design as well. Could incorporate a map of locations as a side panel for the global audience.

**Author Response**

I would update the image to the updated images shown in 1

[Figure]

Figure 1: a) Schematic of the model setup. The black lines indicate the 1D setup where GOTM drives the ECOSMO E2E Ecosystem model and MERCY V2.0 Hg speciation model. These are used to simulate a base case and four scenarios with varying Hg–ecosystem interactions. The impact of the ecosystem is evaluated by comparing the base case to a scenario without: bioaccumulation (Scenario A), bioaccumulation and biogenic reduction (Scenario B), bioaccumulation and partitioning to detritus and DOM (Scenario C), and all mentioned ecosystem interactions (Scenario D).The purple lines show the 3D setup, where the HAMSOM model drives ECOSMO E2E and MERCY V2.0 models. The base case, Scenario A, and Scenario B are simulated in the 3D setup. b) Global map with the regional domain highlighted. c) Regional map of the North and Baltic Sea region. The 3D HAMSOM-ECOSMO-Mercy model domain is marked in blue. The three 1D setups, Northern North Sea (NNS), Southern North Sea (SNS), and Gotland Deep (GD), are labeled and marked with red points.

**Reviewer Comment**

Section 2.4: Include grid resolution for the 3D models (may have missed this)

**Author Response**

That was indeed not specified in this paper but rather only in the original paper. I would update section 2.4 by adding:

**Suggested edit**

The 3D HAMSOM-ECOSMO-MERCY domain covers the Baltic Sea and the North Sea with open boundaries at the English Channel and at 63°N, where the North Sea is connected to the Atlantic Ocean, as shown in Fig. 1. The resolution of the model is about 10x10 km2 on a spherical grid with vertical resolution of 20 layers. The upper four layers are 5 m thick, while the deepest layer reaches a thickness of up to 250 m. The maximum water depth is 630 m.

**Reviewer Comment**

L299: pre-dated?

**Author Response**

That is indeed wrong and should note have the -. Corrected it to:

**Suggested edit**

Fish 2 is at the top of the food chain and is therefore not predated upon in the model.

**Reviewer Comment**

L312-316: A bit more detail on this model tuning/calibration process — what informed the choice of lowered value

**Author Response**

I would suggest adding the following part at the end of section 2.6, replacing the part from line 316 onward with the expanded explanation below:

**Suggested edit**

An essential component to estimate bioaccumulation is the uptake efficiency of carbon, known as assimulation efficiency. Biomagnification occurs if the organic material is absorbed less efficiently or respirated more efficiently than a pollutant, as this would result in an increase in this pollutant compared to organic material in the organism compared to its diet. The assimilation of carbon can be seen as two components; the first absorption refers to all carbon that is used by the fish and not directly excreted via feaces, whereas the assimilation refers to the carbon that is build up into the tissue of the fish. This is investigated for fish in Shelley and Johnson (2022). They found that while the fish have an absorption efficiency of 91-92% they only have an assimilation efficiency of 30-49%. Due to uncertainty, we parameterized the higher trohic-level fish with a lower assimilation efficiency than in the previously published ECOSMO E2E version, down to 45% in fish 2. This was done to tune the model to better reproduce higher $MMHg^+$ bioaccumulation, which is in line with observations. These interactions remain uncertain in the model, but replicating bioaccumulated concentrations are essential to estimate the bioaccumulation feedback on Hg speciation, which is the core focus of this study.

**Reviewer Comment**

L468-472: What are the observed values for biomass? Not sure if I missed their reporting somewhere. Could they also be put on Figure 2 for comparison?

**Author Response**

I would suggest making several changes and changing Fig. 2 by showing chlorophyll-a in the surface water, rather than the fully depth intergrated values. Most measurements measure the concentrations, and hence showing this increases the comparibility to observations. Additionally, I would suggest making the small change to Fig. 2 to make it not based on the last year of the observations but the daily mean of the last 10 years of the simulation to remove the change the plots are influenced by outliers and give a better overview of the behavior of the model. I think the comparison between the model and observations is a bit too nuanced to allow an easy integration of the results into Fig. 2. Because of this, I would suggest adding the Table 1. Then I would add the update Section 3.1 to:

**Suggested edit**

[revised manuscript text omitted]

**Reviewer Comment**

L508: How is "high quality" defined?

**Author Response**

That is indeed badly phrased. There are many more studies in the Baltic Sea that analyze Hg cycling than in the North Sea. Because of this, I suggest reframing this to:

**Suggested edit**

The Baltic Sea is studied more extensively for Hg cycling, and articles such as Kuss (2014) provide the opportunity to validate Hg cycling, while studies such as Nfon et al. (2009) allow the validation of low-trophic-level biota, while data on Hg cycling and bioaccumulation in low-trophic-level biota are extremely limited in the North Sea. Because of this, we focus on the evaluation of bioaccumulation of Hg in the Baltic Sea.

[Figure]

Figure 2: Modeled chlorophyll concentration (left) and organic matter concentration (right). The daily average values over are shown averages over the last 10 years of the simulation (Jan 2001 to Jan 2011). All living organic material is stacked, and detritus and DOM are plotted in the black line on top. Peak spring bloom chlorophyll concentration varies with location. Gotland Deep (a) has 2.2 mg m$^{-3}$ chlorophyll with succession from diatoms to flagellates to cyanobacteria. The Northern North Sea (c) has 5.6 mg m$^{-3}$ chlorophyll and is dominated by flagellates while the Southern North Sea (e) has 4.8 mg m$^{-3}$ chlorophyll and is initially dominated by diatoms and later succeeded by flagellates. All locations have a succession of zooplankton after phytoplankton which microzooplankton and is taken over by mesozooplankton. Fish biomass is stable in the Northern North Sea (fish 1: 1.8-2.6, fish 2: 0.044-0.054 g C m$^{-2}$), the Southern North Sea (fish 1: 2.0-2.2, fish 2: 0.14-0.16 g C m$^{-2}$), and the Gotland Deep (fish 1: 3.0-4.3, fish 2: 0.40-0.44 g C m$^{-2}$). Macrobenthos biomass fluctuates seasonally: Northern North Sea (0.050-2.5 g C m$^{-2}$), Southern North Sea (0.64-13.8 g C m$^{-2}$), while macrobenthos is absent in Gotland Deep due to anoxic bottom water.

---

## Author Comment (AC2)

Answers to reviewer 2

**Reviewer Comment**

Line 5: I find this expression peculiar "Our results show that bioaccumulation can increase total methylmercury (tMeHg) in coastal pelagic waters from 0.059 to 0.092pM, a 44% increase.". Bioaccumulation is a process enhancing concentrations in biota, not in water, which you also state in lines 138-139.

**Author Response**

I would suggest to refrase it as below, that it is clear that the increase is the difference in tMeHg between model runs with and without bioaccumulation.

**Suggested edit**

Incorporating bioaccumulation into the model leads to a 44% increase in total methylmercury (tMeHg) concentrations in coastal pelagic waters, from 0.059 to 0.092pM, compared to a model without bioaccumulation.

**Reviewer Comment**

Line 31: Regarding "This can lead to insufficient data to understand the cycling and bioaccumulation of marine Hg at the base of the food web,": I would argue that undoubtedly, measuring Hg only in biota is insufficient to understand Hg cycling. It may be useful to monitor the ultimate effectiveness of the Minamata treaty but certainly not to understand the observations. I suggest to reformulate this discussion.

**Author Response**

I would refrase that as below:

**Suggested edit**

While measuring Hg in biota can help evaluate the risk of MMHg$^+$ pollution to humans, and thus support the effective evaulation of the minamata convention, fully understanding Hg cycling requires further research. Understanding the link between Hg in the atmosphere and the risk posed to humans by MMHg$^+$ in fish requires studying the factors linking this, including the link between marine Hg cycling and the bioaccumulation of MMHg$^+$ at the base of the food web.

**Reviewer Comment**

Line 41-42: I suggest to replace the work equilibrium since the Hg2+ reduction and Hg0 oxidation are largely mediated by different, independent mechanisms including photochemical and biotic processes.
Line 43: replace the term "double methylated DMHg" with "dimethylmercury DMHg"

**Author Response**

I would suggest to edit line 41-43 as below to address both of the above comments:

**Suggested edit**

The dominant species of Hg in surface water is inorganic $Hg^{2+}$. $Hg^{2+}$ and Hg0 are in dynamic redox cycling, but in aquatic environments this favors the oxidized form, $Hg^{2+}$. Although $Hg^0$ can evaporate, $Hg^{2+}$ can be methylated into two forms of organic Hg, monomethyl mercury ($MMHg^+$) and dimethylmercury (DMHg).

**Reviewer Comment**

Line 70: The discussion on Hg uptake mechanisms here do not harmonize with he discussion of passive diffusion uptake in line 49.

**Author Response**

I see the conflict indeed. I would suggest removing the link between chloride and bioaccumulation in line 49 and merging to two observations in line 70 as follows:

**Suggested edit**

As mentioned before, the dominant form of dissolved $Hg^{2+}$ and $MMHg^+$ is $HgCl_2$ and MMHgCl. These compounds can diffuse through cell membranes due to their lipophilic nature or bind to organic matter (Zhong & Wang, 2009). Recent work has expanded on this basic understanding of bioaccumulation and has shown that while these lipophilic compounds can diffuse through the cell membrane, total uptake into phytoplankton is a complex two-step process in which Hg binds first to the phycosphere before it is absorbed into the cell. Recent data suggests that $MMHg^+$ uptake is influenced by cell-dependent factors such as phycosphere thickness and availability of transmembrane channels for $MMHg^+$ transport, while this is not the case for $Hg^{2+}$ (Garcia-Arevalo et al., 2024). This suggests that $Hg^{2+}$ only bioaccumulates due to its lipophilic nature, whereas $MMHg^+$ both bioaccumulates due to its lipophilic nature and is actively transported by the cell.

**Reviewer Comment**

Line 15: not only by "marine" microorganisms.

**Author Response**

I would suggest to update that sentence by removing marine microorganisms, it is indeed correct that also other processes can form MeHg.

**Suggested edit**

Mercury (Hg) is a naturally occurring toxic element that is extremely persistent in the environment (Driscoll et al., 2013). It can be methylated to methylmercury (MeHg), a dangerous neurotoxin that can bioaccumulate in the marine food chain (Mason et al., 1995; Trevors, 1986).

**Reviewer Comment**

Line 34: I suggest to replace "are a perfect tool to" (which is hardly true) with "are an important tool to".

**Author Response**

I fully agree that they are not perfect tool and it is better to say an important tool.

**Suggested edit**

This can lead to insufficient data to understand the cycling and bioaccumulation of marine Hg at the base of the food web, although these processes are essential in linking Hg emissions to (Me)Hg concentrations in seafood. Modeling studies are an important tool to improve our understanding of these complex interactions and can help evaluate the effectiveness of Hg reduction strategies.

**Reviewer Comment**

Line 35: "Because MeHg formation and subsequent bioaccumulation in seafood are the dominant source of Hg exposure to humans,... ". The sentence is grammatically incorrect, formation and bioaccumulation are not sources, seafood is the source.

**Author Response**

I would suggest to update the sentence as below:

**Suggested edit**

Because seafood consumption is the dominant source of Hg exposure in humans, due to the formation and subsequent bioaccumulation of MeHg in marine organisms, Hg levels in the world's oceans are of special concern.

**Reviewer Comment**

Line 45: the following statement is grammatically incorrect: "Since only MMHg+ bioaccumulates, the term MeHg, in this paper, refers to the total methylated fraction of Hg in seawater.".

**Author Response**

I suggest to rewrite as shown below:

**Suggested edit**

In this paper, MeHg refers to all methylated Hg in seawater, this includes both $MMHg^+$ and DMHg. Of these two Hg species, only $MMHg^+$ is known to bioaccumulate.

**Reviewer Comment**

Line 48: replace "inorganic chlorine complexes" with "inorganic chloride complexes".

**Suggested edit**

In seawater, the abundance of chloride ions causes $Hg^{2+}$ and $MMHg^+$ exist mainly in the form of inorganic chloride complexes.

**Reviewer Comment**

Line 54: avoid using the term "species" for microorganisms as it can be confounded with chemical species (which is discussed in the preceding lines).

**Author Response**

That is indeed confusing. I would replace the biological terms throughout the paper, and I would suggest updating line 54 as follows:

**Suggested edit**

Bioaccumulation of Hg occurs when biota take up Hg at a rate higher than that at which it is excreted (Bryan, 1979).

**Author Response**

I would update the term species in line 66 with animals, so it becomes the sentence below:

**Suggested edit**

Biomagnification can be estimated in nature by sampling stable carbon and nitrogen isotopes with Hg to assess both the Hg content and the trophic position of a series of animals (Lavoie et al., 2013).

**Author Response**

In line 79 I would replace species with genera:

**Suggested edit**

Their research showed that certain genera of cyanobacteria in the Baltic Sea (notable Synechococcus and Aphanizomenon) can also react with Hg by reducing dissolved $Hg^{2+}$ to dissolved gaseous $Hg^0$.

**Author Response**

In line 209, 212, 296, 320, 343, 460 and Table 2 I refer to phytoplankton and fish species, here I would replace species in both the captian, the table and the text with taxa.

**Suggested edit**

(Line 209) Diatoms can dominate at the start of the bloom, but other phytoplankton taxa take over once the silicate is depleted.

**Suggested edit**

(296) It is mainly representative of large demersal fish, such as cod (*Gadus spp.*), but would as a functional group also include other large benthic taxa such as whiting (*Micromesistius poutassou*) or haddock (*Melanogrammus aeglefinus*).

**Suggested edit**

(Caption Table 2) Dimensions, shape, and maximum growth and mortality rates of most common phytoplankton taxa in the North and Baltic Seas, to resemble ECOSMO E2E functional groups and the conversion ratio of mg C to $cm^2$ cell membrane and $dm^3$ cell volume.

**Suggested edit**

(Line 320) The surface area is estimated from the most common phytoplankton taxa in the three phytoplankton functional groups for the North and Baltic Seas. The taxa and dimensions are shown in Table 2.

**Suggested edit**

(Line 343)The taxa representing phytoplankton species with a smaller size and therefore a higher uptake rate also have a high $Hg^{2+}$ release rate.

**Suggested edit**

(Line 460) During the autumn, cyanobacteria can become the dominant taxa with a biomass of up to 50 mg C m$^{-3}$, but there is a large variety in the intensity of the bloom and the relative importance of different taxa (Hjerne et al., 2019).

**Author Response**

In line 207, 278, and 380 I can remove the word species while leaving the rest of the sentence with the intended meaning:

**Suggested edit**

(Line 207) The constant mixing allows macrobenthos to feed directly from the phytoplankton bloom, leading to a high macrobenthos stock (Heip et al., 1992). 41.5 m is also deep enough to support larger fish, such as herring and cod.

**Suggested edit**

(Line 278) This means that certain interactions of the marine ecosystem that could biomagnify Hg, such as predation of organisms within the same functional group or even cannibalism, which do not alter nutrient fluxes or organic matter stocks, are not explicitly specified in the model (Arrhenius & Hansson, 1996; Montagnes & Fenton, 2012; Schrum et al., 2006).

**Suggested edit**

(Line 380) The fish rates are based on a study investigating the uptake, release, and turnover rates in the Indo-Pacific fish Harry hotlips *Plectorhinchus gibbosus* (Wang & Wong, 2003).

**Author Response**

In line 377 i refer to the species of Daphnia Pulex. I would replace that as:

**Suggested edit**

This organism is abundant in the Baltic Sea, but not in the North Sea.

**Author Response**

In line 700 I mention cannabilism within the same species. I would rewrite that mentioning it is within the same functional group:

**Suggested edit**

It is important to note that we wanted to implement realistic bioconcentration and trophic transfer rates to not over-tune the model. Several interactions, such as, for example, cannibalism within the functional group, can increase bioaccumulation in ways that are not captured by the model, resulting in both increased bioaccumulation and trophic levels.

**Reviewer Comment**

Line 65: "Biomagnification can be estimated in nature by sampling stable carbon and nitrogen isotopes with Hg" – isotopes and Hg are not sampled, they are measured.

**Suggested edit**

Biomagnification can be estimated in nature by measuring stable carbon and nitrogen isotopes with Hg to assess both the Hg content and the trophic position of a series of species (Lavoie et al., 2013).

**Reviewer Comment**

Line 80-81: avoid to use "reduce" both for chemical reduction and decrease , in particular in the same discussion.

**Author Response**

Thank you, I replaced all mentions of reduced with decreased in the manuscript where appropriate. That is, in lines 80-81 as shown below. Additionally, I would replace the word reduce by decreased, in the abstract, in line 22, 281, 283, 493, 570, 583, 637, 640, 2x in the caption Fig. 9, 664, 665, 675, 682, 683, 685, 707, 2x in the caption of Fig. 13, 712. This way it is consistently clear that I refer to a chemical reduction or a decrease.

**Suggested edit**

Their research showed that certain species of cyanobacteria in the Baltic Sea (notable Synechococcus and Aphanizomenon) can also react with Hg by reducing dissolved $Hg^{2+}$ to dissolved gaseous $Hg^0$. Since $Hg^0$ is volatile and can evaporate, increasing the fraction of $Hg^0$ can decrease the Hg. This process is referred to as biogenic reduction.

**Reviewer Comment**

Lines 138-139: an element cannot "undergo speciation". Speciation is not a process it is the distribution of an element among different chemical forms. An element may undergo changes in speciation.

**Author Response**

I see that I indeed used this term incorreclty, I would suggest to update this as below:

**Suggested edit**

When Hg is bioaccumulated, it can no longer evaporate, undergo photolysis, or participate in chemical reactions that change its speciation. Instead, it is transported with the organism that accumulated it.

**Author Response**

I also update that in the sentence in line 139 and update it as below:

**Suggested edit**

We are interested in this because the bioaccumulation of Hg removes aquatic $Hg^{2+}$ and $MMHg^+$, which can otherwise participate in chemical reactions that alter their speciation.

---

## Author Response (AR1)

**1  Answers to revieuwer 1**

**Author Response**

Dear reviewer,
Thank you very much for the time to review this manuscript and the great suggestions you made. Below is a an overvieuw on the implementation in the paper to adres your suggestions.

**1.1  Mayor comments**

**Reviewer Comment**

1) While I recognize that this is a complex model and study design, the manuscript is on the long end, and there may be opportunities to streamline the text (particularly in 2.2-2.7) to avoid repetition and allow the core messages to come through more clearly. The authors could consider using a Supplemental Information Section to present some details of the model assumptions and parameterization, as well as for some supporting figures (e.g., Figure 12). In addition, the manuscript would benefit from additional review for typos and readability. Some have been flagged below in minor comments.

**Author Response**

Thank you for your insights. We moved the model specifics such as the exact parameterization to the appendix and addititional visualization to the supplement section. Table 2 and Table 3 are moved to the appendix. It is essential to show the exact parameterization used, but the large tables distract from the core message of the paper. Figures 4, 5, 6, and 12 are moved to the supplement. We decided this as they do not contribute to the core message of the paper but showing more detailed model output might be valiable for a reader interested in more a more detailed understanding of the seasonal progression. We also goes the paper an additional proofreading and shorted some segments by removing some repetitive parts.

**Reviewer Comment**

2) It could be helpful to provide a brief summary of the drivers of spatial and temporal variation in the results, as some of these details may be contained in the cited original model papers and therefore less clear to a reader. For example, for seasonality: to what extent is temperature dependence also considered in the bioaccumulation and toxicokinetic modeling, in addition to biomass modeling? For spatial variation: What determines the spatial distribution of higher trophic levels? Is migration relevant and, if so, how is it considered? If not, what additional implications could this have for the spatial dynamics?

**Author Response**

Thank you for your comment. We expanded on this in several ways. I will first discuss in the MERCY and ECOSMO sections respectively what spatial and temperol drivers control biomass and Hg speciation and then add a segment to the discussion. In section 2.2.3 we added:

**Implementation**

[revised manuscript text omitted]

**Reviewer Comment**

3) See below for some places where clarification of some methodological details could be beneficial, potentially in supporting material (e.g., in model-obs comparison for 1D, initial conditions).

**Author Response**

We went over the detailed comments and implemented that. In addition we expanded the model evaluation segment to discuss this better. Additionally, now that the paper is accepted with minor revision we will also direclty publish the full model code including workable setups that include yaml files that can be used to replicate the exact output. We also added the following description of the initial conditions to the model description part:

**Implementation**

Line 477

**Initial conditions**

The model is initialised with uniform conditions throughout the water. The model was initialised with low biomass of 0.1 mgC m$^{-1}$ for all phytoplankton, 0.01 mgC m$^{-1}$ for zooplankton, and 8.0E-5 for fish 1 and 8.0E-6 for fish 2. The model was spun up for 10 years to allow the model to simulate biological fluxes, Hg cycling, and bioaccumulation to stabilise. A difference between the stations was that initial conditions for nutrients of 7.5 $\mu$M nitrate and 0.47 $\mu$M phosphate were initialized in the Northern North Sea, while in the Baltic Sea .5 $\mu$M nitrate and 0.31 $\mu$M phosphate were initialised to better represent local conditions. The exact initial conditions used for all three setups are available via the YAML files that are provided and can be used to replicate the model output.

**Reviewer Comment**

4) The authors may have the opportunity to deepen the reflection on next steps and future directions, given the importance of the call to better represent ecosystem effects in models. Some questions I am particularly curious to get their thoughts on are: a) is model coupling the only way to do this, is it reasonable to do a back-of-the-envelope adjustment factor that is regionally specific; b) how much trophodynamic complexity is needed — does capturing the base of the food web get most of the effect or do fish 1 and 2 shift the patterns; if so, what might be missing in this current simplified representation of the ecosystem

**Author Response**

We added the following part to the discussion segment

**Implementation**

Line 792

**The required ecosystem complexity to capture Hg dynamics**

As discussed, a main conclusion of this paper is showing that the ecosystem is an integral part of Hg cycling, and that this should not be overlooked. However, there is nuance in how the ecosystem should be implemented in Hg cycling models, and there is a trade-off between keeping the model simple and ensuring key drivers are implemented.

**High trophic levels as a reservoir of MeHg**

In most marine ecosystems, the annual average biomass of primary producers is relatively low; rather, there is a very high turnover rate of primary producers during the bloom period. While the exact numbers vary depending on the location and seasonality, high trophic levels can make up a major component of the total ecosystem biomass, especially in winter. As high trophic levels have the most $MMHg^+$ per biomass, they can form a major reservoir of $MMHg^+$. Our model, however, shows that this does not have a major effect on the tHg concentration. This indicates that the inclusion of high trophic levels such as fish might be necessary to correctly estimate the $MMHg^+$ budget, but the inclusion of fish is not necessary to correctly model tHg fluxes. One point of uncertainty here is that this conclusion is based on our implementation of the ecosystem. As discussed in the model limitation segment, several drivers, such as fish migration and the transport of Hg to deep water by sinking carcasses, are not accounted for, and these drivers could still prove to be an essential component of Hg cycling.

**Bentho pelagic coupling**

A key component where the ecosystem is essential for a correct understanding of Hg cycling is the bentho-pelagic coupling. In coastal areas, the consumption of pelagic detritus and Hg bound to it by macrobenthos can be a major flux of organic carbon from the pelagic to the benthic system. The sediment is identified as a key area for Hg methylation; this increased transport of $Hg^{2+}$ from the pelagic to the benthic due to biotic consumption of detritus would constitute a source of $MMHg^+$. Of course, in some areas, sediment is not resuspended, and then increased transport of Hg to the sediment can result in additional burial of Hg. As the macrobenthic influence on the bentho-pelagic coupling, the burial and resuspension are spatially and temporally variable; this cannot be accounted for by a "back of the envelope" estimation. Rather, the inclusion of a realistic bentho-pelagic coupling is essential for Hg speciation models in coastal areas.

**key biota for Hg cycling**

In an important aspect of estimating the role of the ecosystem in Hg cycling is understanding that not all biota affect Hg cycling in a similar way. The clearest of the ecosystem interactions with Hg cycling is the removal of Hg by biota when biomass is high and the release when biomass is low. This is extensively analyzed in this study. However, several biota might have an unexpectedly high impact on Hg cycling. The first example of this, which is also evaluated in this paper, is Baltic Sea cyanobacteria, which can facilitate biogenic reduction. But beyond this, it is demonstrated that some animals of zooplankton and cephalopods can have *in vivo* Hg speciation (Gente et al., 2023; Gorokhova et al., 2020). Another example is that sponges are demonstrated to have very high inorganic Hg levels, suggesting an important role in the bentho-pelagic coupling (Orani et al., 2020).

**1.6   Detailed comments**

**Reviewer Comment**

L22: Number of parties now exceeds the number of signatories (over 150), so could update the number https://minamataconvention.org/en/parties

**Author Response**

The link provided still states that indeed has 152 parties we updated the statement as below to show both:

**Implementation**

Line 21
Due to the consumption of polluted marine wildlife, more than 1000 people died, and more were permanently disabled (Harada, 1995). Efforts to control Hg emissions culminated in the Minamata Convention on Mercury, which is a pledge to reduce Hg emissions(Outridge et al., 2018). It has 152 parties and is currently signed by 128 countries.

**Reviewer Comment**

L52: "In summary, there are three fractions... in our model." Read as confusing as the model hasn't been introduced yet.

**Author Response**

I agree that this is incorrect to refer to this in the model before the model is introduced. I also think the rest of the alinea could be clearer. It is rewritten as below:

**Implementation**

Line 50
Under anoxic conditions, $Hg^{2+}$ binds with $S^{2-}$ to form cinnabar (HgS), which is considered a sink due to its low solubility (Oliveri et al., 2016). In seawater, the abundance of chloride ions causes $Hg^{2+}$ and $MMHg^+$ to exist mainly in the form of inorganic chloride complexes. The speciation of Hg with organic carbon in the marine ecosystem, such as detritus and DOM, is a complex interaction that can influence the speciation, solubility, mobility, membrane permeability, and toxicity of Hg (Ravichandran, 2004). In this study, we refer to three distinct fractions of both $H^{2+}$ and $MMHg^+$: 1) dissolved species not bound to organic material, including species such as $HgCl_2$ and MMHgCl, collectively referred to as $Hg^{2+}$ and $MMHg^+$, 2) $Hg^{2+}$ and $MMHg^+$ bound to dissolved organic matter (DOM), referred to as Hg-DOM and MMHg-DOM, and $Hg^{2+}$ and $MMHg^+$ bound to detritus, referred to as Hg-detritus and MMHg-detritus.

**Reviewer Comment**

L137: As defined in the first sentence, isn't this bioconcentration only?

**Author Response**

That is indeed badly phrased. It changed as below:

**Implementation**

Line 145
*Bioaccumulation* is the increase in $Hg^{2+}$ or $MMHg^+$ in the biota relative to the concentration of the surrounding water.

**Reviewer Comment**

Fig. 1: Typos in title and Scenario C. Could consider overlaying the 1-D vs 3-D component too so that it captures that aspect of the design as well. Could incorporate a map of locations as a side panel for the global audience.

**Author Response**

I would update the image to the updated images shown in 1 Line 197

[Figure]

Figure 1: a) Schematic of the model setup. The black lines indicate the 1D setup where GOTM drives the ECOSMO E2E Ecosystem model and MERCY V2.0 Hg speciation model. These are used to simulate a base case and four scenarios with varying Hg–ecosystem interactions. The impact of the ecosystem is evaluated by comparing the base case to a scenario without: bioaccumulation (Scenario A), bioaccumulation and biogenic reduction (Scenario B), bioaccumulation and partitioning to detritus and DOM (Scenario C), and all mentioned ecosystem interactions (Scenario D).The purple lines show the 3D setup, where the HAMSOM model drives ECOSMO E2E and MERCY V2.0 models. The base case, Scenario A, and Scenario B are simulated in the 3D setup. b) Global map with the regional domain highlighted. c) Regional map of the North and Baltic Sea region. The 3D HAMSOM-ECOSMO-Mercy model domain is marked in blue. The three 1D setups, Northern North Sea (NNS), Southern North Sea (SNS), and Gotland Deep (GD), are labeled and marked with red points.

**Reviewer Comment**

Section 2.4: Include grid resolution for the 3D models (may have missed this)

**Author Response**

That was indeed not specified in this paper but only in the original paper. Section 2.4 i updated as below:

**Implementation**

Line 284

The 3D HAMSOM-ECOSMO-MERCY domain covers the Baltic Sea and the North Sea with open boundaries at the English Channel and at 63°N, where the North Sea is connected to the Atlantic Ocean, as shown in Fig. 1. The resolution of the model is about 10 x 10 km$^2$ on a spherical grid with a vertical resolution of 20 layers. The upper four layers are 5 m thick, while the deepest layer reaches a thickness of up to 250 m. The maximum water depth is 630 m. The first 4 years are used as spin-up and the final year is used for the analyses. The model is run in its default setup, without bioaccumulation or biogenic reduction. The effect of bioaccumulation on both tHg and tMeHg is visualized by plotting the relative difference in tHg and tMeHg caused by the ecosystem. The data is visualized using the cartopy package in Python version 3.11.2.0.

**Reviewer Comment**

L299: pre-dated?

**Author Response**

That is indeed wrong and should note have the -. Corrected it to:

**Implementation**

Line 381
Fish 2 is at the top of the food chain and is therefore not predated upon in the model.

**Reviewer Comment**

L312-316: A bit more detail on this model tuning/calibration process — what informed the choice of lowered value

**Author Response**

We added the following part at the end of section 2.6, replacing the part from line 316 onward with the expanded explanation below:

**Implementation**

Line 401

The modeled trophic levels are shown in Table 2. In the Gotland, Deep macrobenthos is absent because of the anoxic conditions. Except for fish 2 in the Northern North Sea, all functional groups have trophic levels that are lower than observed in the North and Baltic Seas. This was compensated for by reparameterizing the uptake efficiency of carbon used in the previous version of the ECOSMO E2E model. The uptake efficiency of carbon, known as assimilation efficiency, is a key parameter for biomagnification. Biomagnification occurs if the organic material is absorbed less efficiently or respired more efficiently than a pollutant, as this would result in an increase in this pollutant compared to organic material in the organism compared to its diet. The assimilation of carbon can be seen as two components; the first absorption refers to all carbon that is used by the fish and not directly excreted via faeces, whereas the assimilation refers to the carbon that is built up into the tissue of the fish. Fish are typically shown to have a 91-92% absorption efficiency, but only a 30-49% assimilation efficiency (Shelley & Johnson, 2022).

Due to uncertainty, we parameterized the higher trophic-level fish with a lower assimilation efficiency than in the previously published ECOSMO E2E version, down to 45% in fish 2. To compensate for the decrease in carbon intake, the mortality and respiration rate of zooplankton is decreased to a value that is lower than in Daewel et al. (2019), but still within the range used in previously published models (Cruz et al., 2021). The phytoplankton is parameterized as shown in Table A1. To compensate for the increased zooplankton grazing, the growth rate is increased compared to the previously published ECOSMO E2E version, but remains within the experimentally observed range (Stelmakh & Kovrigina, 2021). All other values are the same as in (Daewel et al., 2019). This was done to tune the model to better reproduce higher $MMHg^+$ bioaccumulation, which is in line with observations. These interactions remain uncertain in the model, but replicating bioaccumulated concentrations is essential to estimate the bioaccumulation feedback on Hg speciation, which is the core focus of this study.

**Reviewer Comment**

L468-472: What are the observed values for biomass? Not sure if I missed their reporting somewhere. Could they also be put on Figure 2 for comparison?

**Author Response**

We updated the model evaluation segment to show this and made some changes to Fig. 2 by showing chlorophyll-a in the surface water, rather than the fully depth intergrated values. Most measurements measure the concentrations, and hence showing this increases the comparibility to observations. Additionally, we changed Fig. 2 to make it not based on the last year of the observations but the daily mean of the last 10 years of the simulation to remove the change the plots are influenced by outliers and give a better overview of the behavior of the model. I think the comparison between the model and observations is a bit too nuanced to allow an easy integration of the results into Fig. 2. Because of this, I would suggest adding the Table 1. Then I would add the update Section 3.1 to:

**Implementation**

Line 494

**Evaluation of carbon fluxes**

[revised manuscript text omitted]

**Author Response**

That is indeed badly phrased. There are many more studies in the Baltic Sea that analyze Hg cycling than in the North Sea. Because of this, we reframed this too:

**Implementation**

Line 563

The modeled Hg bioaccumulation for biota in the Gotland Deep was compared to observations, because the Baltic Sea is studied more extensively for Hg cycling, and studies such as performed by Kuss (2014) provide the opportunity to validate Hg cycling, while studies such as Nfon et al. (2009) allow the validation of low-trophic-level biota, while data on Hg cycling and bioaccumulation in low-trophic-level biota are extremely limited in the North Sea. For this evaluation, we only used model output where the biomass exceeded 100 mg carbon $m^{-2}$ and took the modeled values of the <5 m for phytoplankton and <20 m for other biota to ensure that our bioaccumulation values resemble the modeled values in surface water where biota are active. A comparison between our modeled and observed bioaccumulation is shown in Table 4. Additionally, year-round bioaccumulation of $Hg^{2+}$ and $MMHg^{+}$ in the Gotland Deep, the Northern North Sea, and the Southern North Sea is shown in the supplementary information in Figures S1, S2, and S3 respectively.

**2 Answers to reviewer 2**

**Author Response**

Dear reviewer,
Thank you very much for comments. We answered already how we would suggest to implement your recommendations but after proofreading some formulations have slightly changed. Below we have summarised how we have implemented all your suggestions in the new version of the manuscript.

**Reviewer Comment**

Line 5: I find this expression peculiar "Our results show that bioaccumulation can increase total methylmercury (tMeHg) in coastal pelagic waters from 0.059 to 0.092pM, a 44% increase.". Bioaccumulation is a process enhancing concentrations in biota, not in water, which you also state in lines 138-139.

**Author Response**

This is refrased as below to ensure that it is clear that the increase is the difference in tMeHg between model runs with and without bioaccumulation.

**Implementation**

Line 5
Incorporating bioaccumulation into the model leads to a 44% increase in total methylmercury (tMeHg) concentrations in coastal pelagic waters, from 0.059 to 0.092 pM, compared to a model without bioaccumulation. Bioaccumulation and binding of Hg to organic matter contribute to elevated Hg levels in surface waters.

**Reviewer Comment**

Line 31: Regarding "This can lead to insufficient data to understand the cycling and bioaccumulation of marine Hg at the base of the food web,": I would argue that undoubtedly, measuring Hg only in biota is insufficient to understand Hg cycling. It may be useful to monitor the ultimate effectiveness of the Minamata treaty but certainly not to understand the observations. I suggest to reformulate this discussion.

**Author Response**

This is refrased as below:

**Implementation**

Line 32
While measuring Hg in biota can help evaluate the risk of $MMHg^+$ pollution to humans, and thus support the effective evaluation of the Minamata Convention, fully understanding Hg cycling requires further research. Understanding the link between Hg in the atmosphere and the risk posed to humans by $MMHg^+$ via the consumption of seafood requires studying the factors linking this, including the link between marine Hg cycling and the bioaccumulation of $MMHg^+$ at the base of the food web. Modeling studies are an important tool to improve our understanding of these complex interactions and can help evaluate the effectiveness of Hg reduction strategies.

**Reviewer Comment**

Line 41-42: I suggest to replace the work equilibrium since the Hg2+ reduction and Hg0 oxidation are largely mediated by different, independent mechanisms including photochemical and biotic processes.

Line 43: replace the term "double methylated DMHg" with "dimethylmercury DMHg"

**Author Response**

That is updated as below:

**Implementation**

Line 44

The dominant species of Hg in surface water is inorganic $Hg^{2+}$. $Hg^{2+}$ and $Hg^0$ are in dynamic redox cycling, but in aquatic environments this favors the oxidized form, $Hg^{2+}$. Although $Hg^0$ can evaporate, $Hg^{2+}$ can be methylated into two forms of organic Hg, monomethylmercury ($MMHg^+$) and DMHg.

**Reviewer Comment**

Line 70: The discussion on Hg uptake mechanisms here do not harmonize with he discussion of passive diffusion uptake in line 49.

**Author Response**

I see the conflict indeed. We removed the link between chloride and bioaccumulation in line 49 and merging to two observations in line 70 as follows:

**Implementation**

Line 72

As mentioned before, the dominant form of $Hg^{2+}$ and $MMHg^+$ in the marine environments is $HgCl_2$ and MMHgCl respectively. These compounds can diffuse through cell membranes due to their lipophilic nature or bind to organic matter (Zhong & Wang, 2009). This diffusion is mainly dependent on the surface area of organic membranes that are in contact with water and is therefore dominated by microorganisms such as phytoplankton (Mason et al., 1996). Recent work has expanded on this basic understanding of bioaccumulation and has shown that while these lipophilic compounds can diffuse through the cell membrane, total uptake into phytoplankton is a complex two-step process in which Hg binds first to the phycosphere before it is absorbed into the cell. Recent data suggest that $MMHg^+$ uptake is influenced by cell-dependent factors such as phycosphere thickness and availability of transmembrane channels for $MMHg^+$ transport, while this is not the case for $Hg^{2+}$ (Garcia-Arevalo et al., 2024). This suggests that $Hg^{2+}$ only bioaccumulates due to its lipophilic nature, whereas $MMHg^+$ both bioaccumulates due to its lipophilic nature and is actively transported by the cell.

**Reviewer Comment**

Line 15: not only by "marine" microorganisms.

**Author Response**

The sentence is updated by removing marine microorganisms, it is indeed correct that also other processes can form MeHg.

**Implementation**

Line 15
Mercury (Hg) is a toxic pollutant that poses significant risks to marine ecosystems and human health as a result of bioaccumulation. Despite its known hazards, the processes that govern Hg bioaccumulation within the marine food web are poorly understood. This study examines the role of the marine ecosystem in Hg cycling in highly productive coastal seas.

**Reviewer Comment**

Line 34: I suggest to replace "are a perfect tool to" (which is hardly true) with "are an important tool to".

**Author Response**

I fully agree that they are not perfect tool and it is is updatd as below:

**Implementation**

Line 33
Understanding the link between Hg in the atmosphere and the risk posed to humans by $MMHg^+$ via the consumption of seafood requires studying the factors linking this, including the link between marine Hg cycling and the bioaccumulation of $MMHg^+$ at the base of the food web. Modeling studies are an important tool to improve our understanding of these complex interactions and can help evaluate the effectiveness of Hg reduction strategies.

**Reviewer Comment**

Line 35: "Because MeHg formation and subsequent bioaccumulation in seafood are the dominant source of Hg exposure to humans,... ". The sentence is grammatically incorrect, formation and bioaccumulation are not sources, seafood is the source.

**Author Response**

We updated that sentence as below:

**Implementation**

Line 38
Because seafood consumption is the dominant source of Hg exposure in humans, due to the formation and subsequent bioaccumulation of MeHg in marine organisms, Hg levels in the world's oceans are of special concern.

**Reviewer Comment**

Line 45: the following statement is grammatically incorrect: "Since only MMHg+ bioaccumulates, the term MeHg, in this paper, refers to the total methylated fraction of Hg in seawater.".

**Author Response**

That is rewriten as below:

**Implementation**

Line 48
In this paper, MeHg refers to all methylated Hg in seawater, this includes both $MMHg^+$ and DMHg. Of these two Hg species, only $MMHg^+$ is known to bioaccumulate.

**Reviewer Comment**

Line 48: replace "inorganic chlorine complexes" with "inorganic chloride complexes".

**Implementation**

Line 52
In seawater, the abundance of chloride ions causes $Hg^{2+}$ and $MMHg^+$ to exist mainly in the form of inorganic chloride complexes.

**Reviewer Comment**

Line 54: avoid using the term "species" for microorganisms as it can be confounded with chemical species (which is discussed in the preceding lines).

**Author Response**

That is indeed confusing. We replaced the biological terms as below::

**Implementation**

Line 58
Bioaccumulation of Hg occurs when biota take up Hg at a rate higher than that at which it is excreted (Bryan, 1979).

**Implementation**

Line 67
Biomagnification can be estimated in nature by sampling stable carbon and nitrogen isotopes with Hg to assess both the Hg content and the trophic position of a series of animals (Lavoie et al., 2013).

**Implementation**

Line 87
Their research showed that certain genera of cyanobacteria in the Baltic Sea (notable Synechococcus and Aphanizomenon) can also react with Hg by reducing dissolved $Hg^{2+}$ to dissolved gaseous $Hg^0$.

> **Implementation**
>
> Line 231
> Diatoms can dominate at the start of the bloom, but other phytoplankton taxa take over once the silicate is depleted.

> **Implementation**
>
> Line 379
> It is mainly representative of large demersal fish, such as cod (*Gadus spp.*), but would as a functional group also include other large benthic taxa such as whiting (*Micromesistius poutassou*) or haddock (*Melanogrammus aeglefinus*).

> **Implementation**
>
> (Caption Table 2) Dimensions, shape, and maximum growth and mortality rates of most common phytoplankton taxa in the North and Baltic Seas, to resemble ECOSMO E2E functional groups and the conversion ratio of mg C to $cm^2$ cell membrane and $dm^3$ cell volume.

> **Implementation**
>
> Line 304
> The surface area is estimated from the most common phytoplankton taxa in the three phytoplankton functional groups for the North and Baltic Seas. The taxa and dimensions are shown in Table 2.

> **Implementation**
>
> Line 328
> The groups representing phytoplankton taxa with a smaller size and therefore a higher uptake rate also have a high $Hg^{2+}$ release rate. As a result, all phytoplankton groups reach equilibrium at similar $Hg^{2+}$ concentrations.

> **Implementation**
>
> Line 512
> During the autumn, cyanobacteria can become the dominant taxa with a biomass of up to 50 mg C $m^{-3}$, but there is a large variety in the intensity of the bloom and the relative importance of different taxa (Hjerne et al., 2019).

> **Implementation**
>
> Line 228
> The constant mixing allows macrobenthos to feed directly from the phytoplankton bloom, leading to a high macrobenthos stock (Heip et al., 1992). 41.5 m is also deep enough to support larger fish, such as herring and cod.

> **Implementation**
>
> Line 387
> This means that certain interactions of the marine ecosystem that could biomagnify Hg, such as predation of organisms within the same functional group or even cannibalism,

which do not alter nutrient fluxes or organic matter stocks, are not explicitly specified in the model (Arrhenius & Hansson, 1996; Montagnes & Fenton, 2012; Schrum et al., 2006).

**Implementation**

Line 369
The fish rates are based on a study investigating the uptake, release, and turnover rates in the Indo-Pacific fish Harry hotlips *Plectorhinchus gibbosus* (Wang & Wong, 2003).

**Implementation**

Line 367
Water fleas are abundant in the Baltic Sea, but not in the North Sea.

**Implementation**

Line 835
It is important to note that we wanted to implement realistic bioconcentration and trophic transfer rates to not over-tune the model. Several interactions, such as, for example, cannibalism within the functional group, can increase bioaccumulation in ways that are not captured by the model, resulting in both increased bioaccumulation and trophic levels.

**Reviewer Comment**

Line 65: "Biomagnification can be estimated in nature by sampling stable carbon and nitrogen isotopes with Hg" – isotopes and Hg are not sampled, they are measured.

**Implementation**

Line 67
Biomagnification can be estimated in nature by measuring stable carbon and nitrogen isotopes with Hg to assess both the Hg content and the trophic position of a series of species (Lavoie et al., 2013).

**Reviewer Comment**

Line 80-81: avoid to use "reduce" both for chemical reduction and decrease , in particular in the same discussion.

**Author Response**

Thank you, we replaced all mentions of reduced with decreased in the manuscript where appropriate. That is, in lines 80-81 as shown below. Additionally, I would replace the word reduce by decreased, in the abstract, in line 22, 281, 283, 493, 570, 583, 637, 640, 2x in the caption Fig. 9, 664, 665, 675, 682, 683, 685, 707, 2x in the caption of Fig. 13, 712. This way it is consistently clear that I refer to a chemical reduction or a decrease.

**Implementation**

Line 87
Their research showed that certain species of cyanobacteria in the Baltic Sea (notable Synechococcus and Aphanizomenon) can also react with Hg by reducing dissolved $Hg^{2+}$

> to dissolved gaseous $Hg^0$. Since $Hg^0$ is volatile and can evaporate, increasing the fraction of $Hg^0$ can decrease the Hg. This process is referred to as biogenic reduction.

**Reviewer Comment**

Lines 138-139: an element cannot "undergo speciation". Speciation is not a process it is the distribution of an element among different chemical forms. An element may undergo changes in speciation.

**Author Response**

This is indeed used incorreclty, we update this as below:

**Implementation**

Line 145: When Hg is bioaccumulated, it can no longer evaporate, undergo photolysis, or participate in chemical reactions that change its speciation; instead, it is transported with the organism that accumulated it.

**Author Response**

I also update that in the sentence in line 139 and update it as below:

**Implementation**

[revised manuscript text omitted]

Orani, A. M., Vassileva, E., Azemard, S., & Thomas, O. P. (2020). Comparative study on Hg bioaccumulation and biotransformation in Mediterranean and Atlantic sponge species. *Chemosphere*, *260*, 127515.

OSPAR. (2017). Winter Nutrient Concentrations in the Greater North Sea, Kattegat and Skagerrak.

Outridge, P. M., Mason, R. P., Wang, F., Guerrero, S., & Heimbürger-Boavida, L. E. (2018, October). Updated Global and Oceanic Mercury Budgets for the United Nations Global Mercury Assessment 2018.

Ravichandran, M. (2004). Interactions between mercury and dissolved organic matter - A review. *Chemosphere*, *55*(3), 319–331.

Schrum, C., Alekseeva, I., & St. John, M. (2006). Development of a coupled physical–biological ecosystem model ECOSMO: Part I: Model description and validation for the North Sea. *Journal of Marine Systems*, *61*(1-2), 79–99.

Shelley, C. E., & Johnson, D. W. (2022). Larval fish in a warming ocean: a bioenergetic study of temperature-dependent growth and assimilation efficiency. *Marine Ecology Progress Series*, *691*, 97–114.

Sparholt, H. (1990). An estimate of the total biomass of fish in the North Sea. *J. Cons. int. Explor. Mer*, *46*, 200–210.

Stelmakh, L., & Kovrigina, N. (2021). Phytoplankton Growth Rate and Microzooplankton Grazing under Conditions of Climatic Changes and Anthropogenic Pollution in the Coastal Waters of the Black Sea (Sevastopol Region).

Thurow, F. (1997). Estimation of the total fish biomass in the Baltic Sea during the 20th century. *ICES Journal of Marine Science*, *54*, 444–461.

Wang, W., & Wong, R. (2003). Bioaccumulation kinetics and exposure pathways of inorganic mercury and methylmercury in a marine fish, the sweetlips Plectorhinchus gibbosus. *Marine Ecology Progress Series*, *261*.

Zhong, H., & Wang, W.-X. (2009). Controls of Dissolved Organic Matter and Chloride on Mercury Uptake by a Marine Diatom. *Environ. Sci. Technol.*, *43*(23), 8993–9003.